# Simulating glacial dust changes in the Southern Hemisphere using ECHAM6.3-HAM2.3

Stephan Krätschmer[1], Michèlle van der Does[1], Frank Lamy[1], Gerrit Lohmann[1,2], Christoph Völker[1], Martin Werner[1]

[1]Alfred Wegener Institute, Helmholtz Centre for Polar and Marine Research, Bremerhaven, Germany
[2]Physics Department, University of Bremen, Bremen, Germany

*Correspondence to*: Stephan Krätschmer (stephan.kraetschmer@awi.de)

**Abstract.** Mineral dust aerosol constitutes an important component of the Earth's climate system, not only on short timescales due to direct and indirect influences on the radiation budget, but also on long timescales by acting as a fertilizer for the biosphere and thus affecting the global carbon cycle. For a quantitative assessment of its impact on the global climate, state-of-the-art atmospheric and aerosol models can be utilized. In this study, we use the ECHAM6.3-HAM2.3 model to perform global simulations of the mineral dust cycle for present-day (PD), pre-industrial (PI) and Last Glacial Maximum (LGM) climate conditions. The intercomparison with marine sediment and ice core data as well as other modeling studies shows that the obtained annual dust emissions of 1221, 923 and 5159 Tg for PD, PI and LGM, respectively, generally agree well with previous findings. Our analyses focussing on the Southern Hemisphere suggest that over 90 % of the mineral dust deposited over Antarctica are of Australian or South American origin during both PI and LGM. However, contrary to previous studies, we find that Australia contributes a higher proportion during the LGM, which is mainly caused by changes in the precipitation patterns. Obtained increased particle radii during the LGM can be traced back to increased sulphate condensation on the particle surfaces as a consequence of longer particle lifetimes. The meridional transport of mineral dust from its source regions to the South Pole takes place at different altitudes, depending on the grain size of the dust particles. We find a trend of generally lower transport heights during the LGM compared to PI as a consequence of reduced convection due to colder surfaces, indicating a vertically less extensive Polar Cell.

## 1 Introduction

In the last decades, mineral dust has been identified to play an important role in the climate system due to its various interactions in atmospheric processes (e.g., Maher et al., 2010). The emission of dust particles takes place in arid and semi-arid areas and is controlled by several meteorological factors and surface properties in the source areas, for instance wind speed, soil moisture, the type and amount of vegetation as well as soil composition and the occurrence of other non-erodible elements (Goudie, 2008). Once suspended, especially smaller dust particles can get transported over large distances and distributed by the global atmospheric circulation before they are removed from the atmosphere by sedimentation, turbulent deposition or scavenging. During transport, the dust particles directly influence Earth's radiation budget by scattering and

absorbing short- and longwave radiation depending on particle size and mineralogical composition (Dufresne et al., 2002; Balkanski et al., 2007), which in turn affects the atmospheric stability by altering the vertical temperature profile and relative humidity (Boucher, 2015). Besides these direct effects on the energy budget, mineral dust particles act as cloud condensation and ice nuclei and thus affect optical and other microphysical properties of clouds, which again influences the radiation
budget (Spracklen et al., 2008).

However, the role of mineral dust in the Earth System is by far not limited solely to the radiative energy balance. Depending on the mineralogical composition of the dust particles, they may constitute a very important source of micronutrients for some ecosystems. By acting as a fertilizer for the vegetation and the biosphere in general, mineral dust plays a crucial role also in the carbon cycle and its impact on the global climate. Saharan dust, for instance, has been
identified to represent an important source of phosphorus for the Amazon rainforest, where the soil shows a general depletion (Reichholf, 1986). Another example for an ecosystem depending on the fertilizing effects of dust is the Southern Ocean. On-board incubation experiments showed that the bio-productivity in this region is strictly limited by iron availability (Martin et al., 1990). Based on the results of these experiments, Martin (1990) proposed the so-called iron hypothesis, which states that increased amounts of bioavailable iron were supplied during glacials in that region by enhanced
mineral dust deposition. As a consequence of the atmospheric iron input, phytoplankton blooms emerged binding huge amounts of $CO_2$. After death, large parts of the organic matter sunk down into the deep sea, effectively removing the bound $CO_2$ from the atmosphere and thus contributing significantly to the observed reduction from 280 ppm to well below 200 ppm $CO_2$ during the last glacial period. Estimates based on palaeoceanographic data constrain the upper limit of this effect to 40 ppm, since increased aeolian dust fluxes are only observed once the atmospheric $CO_2$ concentration had already decreased
by 40 to 50 ppm (Martínez-Garcia et al., 2011).

The analysis of ice cores and marine sediments from both hemispheres has provided detailed data on the spatial and temporal variability of dust deposition fluxes, both in total amount as well as particle size distribution, over the last one million years on a global scale and enabled the utilization of mineral dust as a climate proxy. However, the interpretation of these data turns out to be quite challenging due to their multifactorial causes. For instance, it has not been finally resolved
yet whether climatic changes in the source regions (e.g., soil moisture, wind speed, vegetation cover) or changes in the atmosphere (e.g., wind speed, circulation patterns, particle lifetime) were the primary driver causing up to 20 times higher dust fluxes to Antarctica during glacials (Kohfeld and Ridgwell, 2009; Wolff et al., 2010). The situation is similar with the data on particle sizes. Ice core data from Greenland, for instance, show the deposition of significantly coarser dust during cold climates (Steffensen, 1997), which has been interpreted as a consequence of stronger winds transporting coarser
particles over larger distances, or as a result of the appearance of a closer dust source not active during warmer climates (Mahowald et al., 2014). Contrastingly, ice core data from Antarctica indicate the deposition of generally finer particles during glacials, although a regional analysis reveals the deposition of slightly coarser particles in parts of central East Antarctica (Delmonte et al., 2002, 2004). The regional difference is explained by changes in the atmospheric circulation leading to dust transport pathways of different lengths from the sources to the deposition areas. On longer trajectories, a

larger proportion of coarser particles gets removed during transport and the observed particle-size distribution in the deposition area is consequently shifted to finer particles. In order to support or reject such a hypothesis, the unambiguous identification of the dust's provenance is indispensable. Based on measurements of Strontium (Sr) and Neodymium (Nd) isotope ratios, southern South America has been identified as the most likely source of dust found in Antarctic ice cores which has been delivered during the last glacial (Basile et al., 1997). This finding has been supported by the study of Lunt

and Valdes (2002), who found via a back trajectory modeling approach that "[mineral dust] transport from Patagonia [to Antarctica] is much more efficient than transport from both Australia and South Africa", which the authors explain by the southward extension of Patagonia well into "the strong winds over the Southern Ocean". However, it is still challenging to identify minor source contributions in the presence of a predominant dust source (Vallelonga, 2014). For instance, the characteristic isotope ratios of $^{87}Sr/^{86}Sr$ and $^{143}Nd/^{144}Nd$ of mineral dust found in Antarctic ice cores and delivered during

interglacials not only match those of southern South American soil samples, but also match the ratio of soil samples from central and southeast Australia (De Deckker et al., 2010). This finding suggests that transport of Australian dust to Antarctica is generally possible and raises the question if and to what degree Australian dust sources might also have contributed to the total amount of dust found in ice cores during glacials as well as which climate elements caused the according changes.


In this study, we use a state-of-the-art atmospheric general circulation model coupled to an aerosol model in order to provide new global simulations of the dust cycle for different climate conditions. We compare our present-day simulations to results obtained in the scope of the global dust model intercomparison in AeroCom phase I in order to assess the performance of our model. The main focus of our study is, however, a comprehensive and quantitative characterization of

the global dust cycle during the last glacial maximum (LGM, 21 kyr BP), in particular compared to the pre-industrial (PI, 1850-1879 AD) dust cycle. In order to verify our simulation results, we use observational data for a detailed comparison. Our provenance studies focus on the Southern Hemisphere and give new insights on the respective contribution of the three major dust source regions Australia, South Africa and Patagonia to the total dust deposition in the Southern Hemisphere. Finally, we use the combined global and local information on particle sizes and lifetimes as well as precipitation and wind

patterns to draw conclusions concerning the atmospheric circulation in the Southern Hemisphere during the LGM.

## 2 Model description

In this section, a concise overview on the model ECHAM-HAMMOZ used in our study is given. It consists of the atmospheric circulation model ECHAM6.3 coupled to the aerosol model HAM2.3 as well as the model for atmospheric gas-phase chemistry MOZ1.0. The sub-models HAM and MOZ can both be independently switched on and off. In the scope of

our study, we only run the aerosol model and use monthly prescribed oxidant fields. A detailed description of the complete model is given in Schultz et al. (2018).

## 2.1 Atmospheric general circulation model ECHAM6.3

ECHAM6 (subversion ECHAM6.3.02 in this study) is the latest version of an atmospheric general circulation model developed by the Max-Planck Institute in Hamburg (Stevens et al., 2013). In the late 1980's, it had been branched from a model used at the European Center for Medium Range Weather Forecasts (ECMWF). Since then, new numerical schemes and physical processes have been included and released with each upcoming new model version. The model code can be roughly divided into two parts, one consisting of the adiabatic core and the other consisting of a suite of physical parameterizations for diabatic processes on the sub-grid scale.

The adiabatic core uses a mixed finite-differences/spectral discretization approach to solve the according primitive equations for the hydrodynamic variables vorticity and divergence as well as the thermodynamic variables temperature and surface pressure. A spectral-transform method is used to represent horizontal spatial differences (Simmons et al., 1989), applying a triangular truncation using a finite number of modes (e.g., T63). In the vertical, the model is discretized by either 47 or 95 model layers (L47 or L95) on a Lorenz grid up to a model top of 0.01 hPa, corresponding to a height of about 80 km. The implemented numerical scheme for vertical advection conserves potential as well as kinetic energy, pressure gradients are calculated by a scheme conserving angular momentum (Simmons and Jiabin, 1991). For tracer transport, a flux-form semi-Lagrangian scheme on a Gaussian grid is used due to its inherent conservation properties (Lin and Rood, 1996).

As mentioned above, ECHAM6 comes with a comprehensive set of parameterizations. The boundary layer as well as turbulence are parameterized based on an eddy diffusivity approach (Brinkop and Roeckner, 1995). The model generally allows for shallow, mid-level and deep convection, although only one type of convection is allowed per grid cell with a preference for deep convection. Convective clouds and their according transport are calculated by a mass-flux scheme. The sub-grid orography is parameterized according to Lott (1999) in order to account for momentum transport in the boundary layer as well as gravity waves. Sub-grid scale cloudiness is determined by calculating the cloud fraction based on the relative humidity once a threshold value is exceeded (Sundqvist et al., 1989). While the cloud droplet number concentration is only parameterized depending on the altitude in the base model version of ECHAM6, the coupling to the aerosol model HAM2.3 enables now an explicit calculation based on aerosol activation (see Sect. 2.2). The transport of cloud water and ice is prognostically calculated, accounting for adiabatic circulations, phase transitions and also the conversion to large-scale precipitation (Lohmann and Roeckner, 1996). Radiative transfer is parameterized by dividing the relevant part of the electromagnetic spectrum into 14 shortwave and 16 longwave bands, and absorption is calculated over all bands depending on trace gas concentrations and ambient pressure using look-up tables. The optical properties of clouds for each band are calculated based on Mie theory, for aerosols the according treatment is described in section 2.2. A novelty that came up with ECHAM6 is the integration of the land-surface and vegetation model JSBACH (Reick et al., 2013). Therein, each grid cell is assumed to be tiled allowing various shares of bare soil and 12 different plant functional types, whereas the soil hydrology is

represented by a five-layer scheme (Hagemann and Stacke, 2015). The sub-model allows for the calculation of dynamic vegetation and provides physical parameters like evaporation and surface albedo.

## 2.2 Aerosol model HAM2.3

In this subsection, a brief overview on the Hamburg Aerosol Model (HAM) is given. Essentially, it constitutes a comprehensive microphysics package taking into account all relevant processes and interactions of the five atmospheric aerosol species sulphate, black carbon, organic carbon, sea salt and mineral dust. The very first version was developed and coupled to ECHAM5 by Stier et al. (2005). Since then, it has been extended several times by new numerical schemes and considered processes as well as coupled to the latest ECHAM release (Tegen et al., 2019; Neubauer et al., 2019).

### 2.2.1 General overview

Similar to ECHAM6.3, HAM2.3 can also be roughly split up into two parts, with one mainly treating large-scale processes like emission, transport and deposition of aerosol particles, while the other deals with the microphysical processes such as nucleation, condensation, coagulation and hydration. For the latter, the default sub-model in HAM2.3 is the modal M7 aerosol model (Vignati et al., 2004), which represents the particle-size distribution via seven superimposed log-normal modes and is also used in our study.

The seven log-normal modes in M7 consist of four soluble and three insoluble modes, representing the nucleation mode (number median radius $\bar{r} <$ 5nm, only soluble mode), Aitken mode (5nm $< \bar{r} <$ 50nm), accumulation mode (50nm $< \bar{r} <$ 500nm) and coarse mode ($\bar{r} >$ 500nm). Hereby, the soluble modes are assumed to be perfectly internally mixed. Each mode is mathematically defined by three moments of the distribution, namely the aerosol number N, the number median radius $\bar{r}$ and the standard deviation σ. The latter is set to σ = 1.59 for the nucleation, Aitken and accumulation modes and σ = 2.00 for the coarse modes (Wilson et al., 2001). In each time step, HAM calculates the transport of the tracer aerosol mass and aerosol number. Subsequently, the number median radius $\bar{r}$ can be determined for each mode and grid cell based on the given information. Each mode is consequently confined by the boundaries given above, though the number median radius changes over time due to various processes transferring particles between modes, respectively, removing or adding particles from or to modes. All further size-dependent processes are calculated based on this number median radius for all particles of a specific mode. Aerosol particles devolve from insoluble to soluble modes either due to condensation of sulfate on their surface or due to coagulation with particles of soluble modes. The aerosol dynamics is based on a general coagulation equation, calculating the change in the particle number for each mode and time step considering inter- and intra-modal coagulation as well as sources and sinks. Interactions between aerosol particles and clouds are parameterized by an explicit activation scheme based on Köhler theory (Abdul-Razzak and Ghan, 2000). As mentioned above, HAM enables the explicit calculation of cloud droplet and ice crystal number concentrations, which is done via a two-moment cloud microphysics scheme (Lohmann et al., 2007; Lohmann and Hoose, 2009). The optical properties of the aerosols are not calculated online in order to save on computational costs. Instead, a look-up table provides pre-calculated values based on Mie theory and

contains the Mie size-parameter as well as the real and imaginary part of the refractive index. Aerosols and aerosol precursor are emitted from various natural (biosphere, ocean, …) and anthropogenic sectors (industry, ships, power plants, ...), where natural emissions are calculated online while emissions from anthropogenic sectors are provided in form of external input fields (see Sect. 2.2.3). Considered particle removal processes include dry deposition, sedimentation based on a Stokes
settling velocity approach as well as particle size-dependent in- and below-cloud scavenging (Croft et al., 2009, 2010).

### 2.2.2 Treatment of Mineral Dust in HAM2.3

Of particular importance in the scope of this study is the treatment of mineral dust aerosol in HAM2.3. Since dust is only emitted from arid and semi-arid areas without or covered only by low vegetation, the determination of those regions is a crucial point in order to attain a suitable parameterization of the dust emission process. We use a scheme introduced by
170 Stanelle et al. (2014), which enables a fully interactive coupling between JSBACH and ECHAM6 with respect to dust emissions and was developed in order to investigate the impact of anthropogenic land use change on the global dust cycle. Hereby, terrestrial tiles not covered by any vegetation represent potential dust source areas, whereas tiles covered by snow or vegetation block dust emissions. While gaps in low-stature vegetation such as shrubs, crops etc. allow for dust emissions, gaps in forests prevent them. The direct coupling of the dust emissions to the land surface and vegetation model
automatically accounts for any changes, for instance in the land-sea distribution as a consequence of past climate changes.

The dust emission process is parameterized based on the wind speed at 10 m elevation. In order to determine the total emission flux, particles of all soil types are divided into 192 dust size bins ranging from 0.2 μm to 1300 μm and a threshold friction velocity is calculated individually for each bin (Marticorena et al., 1997). The calculation of the saltation process is based on an explicit formulation of Marticorena and Bergametti (1995). Since water increases the cohesive forces
among dust particles, dust emission is prohibited once the soil moisture of the uppermost layer exceeds a threshold value. In all dust source regions, a constant surface roughness length of 0.001 cm is prescribed. The vertical dust emission flux is finally calculated based on the horizontal dust flux according to White (1979), whereas the particles are emitted either into the insoluble accumulation mode (mass median radius (mmr) = 0.37 μm, σ = 1.59) or the insoluble coarse mode (mmr = 1.75 μm, σ = 2.00). Due to their short lifetimes, emissions into the supercoarse mode are neglected (Stier et al., 2005; Cheng
et al., 2008). Since the surface orography is not taken into account in ECHAM6.3 in order to determine the aerodynamic surface roughness length, so-called regional correction factors are applied on the threshold friction velocity to account for "subsequent changes in surface wind distributions over dust source areas" (Tegen et al., 2019). They can be set for each dust source region individually and are chosen such that the simulated emissions match best with values by Huneeus et al. (2011). We also use those factors in the scope of our study to switch on/off specific dust source areas individually for the according
experiments.

### 2.2.3 Simulation setup and experiments

We perform global simulations for present day (PD, 1996-2005 AD), pre-industrial (PI, 1850-1880 AD) and last glacial maximum (LGM, 21 kyr BP) climate conditions using the spatial resolution T63 (1.875° x 1.875°) with 47 vertical layers. While an initial run ("cold start") is performed for PD and PI, the model is initialized by restart files for the LGM, which represent a dynamic equilibrium of the model obtained after several hundred simulation years for the according topographic, vegetation and climate conditions. The spin-up time is 10 years for PD and PI and 20 years for the LGM, and the total simulation period is 20 years for PD, 40 years for PI and 50 years for LGM. Except for PD, the final 30 simulation years are evaluated in order to calculate mean values. Our simulation setup consists of an atmosphere-only model, i.e., our model is not coupled to an ocean model. Instead, we provide the boundary conditions for the sea surface temperature as well as the sea ice concentration in form of monthly-resolved input files. For PD, we use monthly resolved 30-year means for the years 1979-2008 based on the Atmospheric Model Intercomparison Project – Phase II (AMIP II) dataset (Taylor et al., 2000), for PI we prescribe monthly-resolved 30-year means for the years 1870-1899 created in the scope of the Program for Climate Model Diagnosis and Intercomparison (PCMDI) based on the latest AMIP II dataset (Durack and Taylor, 2019) and for the LGM we use monthly-resolved 30-year means based on reconstructions in the scope of the Glacial Atlantic Ocean Mapping (GLAMAP, Paul and Schäfer-Neth, 2003). The emission of aerosols and aerosol precursor from various anthropogenic sectors (see Schultz et al. (2018)) for PI and PD are prescribed in form of monthly-resolved input files based on the Atmospheric Chemistry and Climate Model Intercomparison Project (ACCMIP) dataset (Lamarque et al., 2010). For LGM runs, greenhouse gas concentrations as well as orbital parameters have been set in accordance with the Paleoclimate Model Intercomparison Project - Phase 4 (PMIP4) experimental setup (Kageyama et al., 2017). Since a large amount of water was bound in ice sheets covering large parts of North America and northern Europe during the LGM, also the land-sea mask has been updated accordingly in order to account for a 125 m lower sea level compared to present-day conditions (e.g., Clark et al., 2009). In all simulations, dynamic vegetation is calculated online by JSBACH. It should be mentioned that the JSBACH restart files used for the LGM run initialize a desert in a small region at the north-eastern coast of South America. Albeit efforts to reconstruct the vegetation in the Amazon region show indeed a patch-like retreat of the rainforest during the LGM, the surrounding vegetation is rather suggested to have consisted of tropical grassland instead of a desert (e.g., Ray and Adams (2001)). Consequently, we prohibit dust emissions in this region and accept this small shortcoming of the land-surface and vegetation model.

## 3. Results and discussion

### 3.1 Model performance for present-day climate conditions

The present-day simulation is intended to evaluate the model's performance concerning the representation of the global dust cycle. Table 1 shows a comparison of several key values used to characterize the global dust cycle budget of our modeling results in relation to the global dust model intercomparison in AeroCom phase I for present-day climate conditions (Huneeus et al., 2011). In terms of total annual dust emissions, our model simulates 1221 Tg yr$^{-1}$, which is close to the AEROCOM

median values of 1123 Tg yr$^{-1}$. Generally, our model performs rather at the lower end of the 1000-4000 Tg yr$^{-1}$ dust

emissions estimated by the IPCC (Intergovernmental Panel on Climate Change, 2014), which is caused by the neglect of a supercoarse particle mode (Stanelle et al., 2014). As a consequence of the comparably small particle sizes, the simulated atmospheric dust burden of 19.8 Tg is slightly higher than the AEROCOM median of 15.8 Tg, which also leads to a higher averaged particle lifetime of 5.96 d compared to the 4.59 d for the AEROCOM median. A noticeable difference is also the respective contribution of the three different deposition mechanisms wet deposition, dry deposition and sedimentation. While

the AEROCOM median suggests an almost equal proportion of 357, 396 and 314 Tg yr$^{-1}$, respectively, the most dominant deposition mechanism in our model is wet deposition with 708 Tg yr$^{-1}$, followed by sedimentation (412 Tg yr$^{-1}$) and finally dry deposition (93 Tg yr$^{-1}$). Again, this can be explained with the comparably small particle sizes: dry deposition and sedimentation apply first and foremost to larger particles close to the source regions, while rain-scavenging is the predominant deposition mechanism in remote areas after long-range transport, which mostly applies in particular to smaller

particles. Based on this intercomparison we conclude that the model setup used in our study represents the global dust cycle adequately for the following investigations. A comparison of the simulated total dust deposition with observational data from 84 sites provided by Huneeus et al. (2011) can be found in the supplement (Fig. S1, S2).

## 3.2 The dust cycle under pre-industrial and last glacial maximum climate conditions

### 3.2.1 Overview

Figure 1 shows the results concerning the global dust cycle for PI and LGM. We find an annual global dust emission of 923 and 5159 Tg yr$^{-1}$ for PI and LGM, respectively. Our model identifies North Africa and Asia as major dust source regions in the Northern Hemisphere and southern South America, southern Africa and Australia in the Southern Hemisphere (Fig. 1a, b). The zonally-averaged dust emissions shown in Fig. 1c reveals that while all sources emit noticeably more dust during the LGM compared to PI, the increase is most significant for Asia, North Africa as well as Australia. As a consequence of the

higher dust emissions during the LGM, the dust burden also increases from 16 to 99 Tg. Once emitted, the dust is distributed by the atmospheric circulation (Fig. 1d, e). The north- and southeasterly trade winds transport in particular North and southern African dust along the equator over the Atlantic Ocean, while dust emitted in Asia as well as Australia and southern South America gets transported by the Westerlies over the Pacific Ocean and the Southern Ocean, respectively. Finally, the dust deposition patterns naturally follow the dust burden patterns (Fig. 1g, h). As expected, the depositions in mid-latitudes

close to the source regions are higher during the LGM (Fig. 1i). However, it should be noted that also in remote regions in both hemispheres more dust deposition occurs, in particular in the high latitudes. Our model simulates a deposition of 423 Tg yr$^{-1}$ (45 %) and 2122 Tg yr$^{-1}$ (41 %) onto the oceans globally for PI and LGM.

       With respect to dust emissions, our values are lower than the simulated 2785 Tg yr$^{-1}$ for PI and 6294 Tg yr$^{-1}$ for the LGM by the Community Earth System Model (CESM) (Albani and Mahowald, 2019), whereas the differences are more

significant for PI (Table 2). The major dust source regions we identified for both the Northern and the Southern Hemisphere

are in accordance with other studies (e.g., Takemura et al. (2009)), suggesting that the interactive coupling of dust emissions to the land-surface and vegetation model JSBACH in our model yields reliable results (see Table 3 for detailed regional values). As a consequence of the atmospheric circulation pattern, the increased zonally-averaged dust burden in the high northern latitudes during the LGM (Fig. 1f) can at least partly be attributed to the markedly higher dust emissions in Asia

compared to PI, which is in accordance with results by Werner et al. (2002). With respect to dust deposition, our model simulates a higher relative proportion of dust deposited over the oceans than the 440 Tg yr$^{-1}$ (16 %) for PI and the 826 Tg yr$^{-1}$ (13 %) for LGM found by Albani et al. (2016) using CESM. Shao et al. (2011) evaluated results of several modeling studies and found on average a dust emission of 2000 Tg yr$^{-1}$, of which 25 % deposit over the oceans. Regardless of the differences in absolute numbers it is worth noting that both models indicate a decreased deposition proportion for LGM climate

conditions. However, it is possible that our prescribed SSTs are too warm (Tierney et al., 2020). While our study does not indicate a suggested equatorward shift of the Westerlies, other studies have shown an influence of prescribed SSTs and SIC on the Westerlies (e.g., Sime et al. (2013)).

Based on the previous findings, it can be concluded that the differences of ECHAM6.3-HAM2.3 in comparison to CESM in terms of dust modeling can be traced back to the modeled particle size distribution. The atmospheric component of

CESM, CAM4, utilizes a sectional approach in order to represent the particle size distribution, grouping the particles into 4 size bins ranging from 0.1 to 10 μm in diameter (Mahowald et al., 2006). The higher dust emission fluxes in CESM, in particular for PI climate conditions, are consequently caused by larger particles, which add significantly to the mass budget and their rapid sedimentation close to the source regions lowers the average particle lifetime (Albani et al., 2014). The smaller particle sizes modeled by ECHAM6.3-HAM2.3, however, enable long-range transport to remote regions for a large

proportion of the emitted dust and consequently causes wet scavenging to become the predominant deposition mechanism. The importance of an adequate modeling of finer dust in order to achieve a proper representation of long-range transport and subsequent deposition over the oceans has already been pointed out by Mahowald et al. (2014). Eventually, the question how well our model performs in absolute values can only be evaluated by comparing the results to measurement data, which is done in section 3.2.2.


On average 5.6 times higher dust emissions are simulated for LGM compared to PI climate conditions (Table 3). While the increase is only 5.3-fold in the Northern Hemisphere, the ratio is even 9.8 for the Southern Hemisphere. This raises the question concerning the causes of this significant increase. One obvious reason is the difference in the land-sea distribution, as can be seen in Fig. 1. Globally, the extended drylands during the LGM in the coastal regions as a consequence of the

lower sea level, for instance in South America and Australia (Fig. 1a, b), emit 229 Tg yr$^{-1}$ of mineral dust and thus contribute only around 5 % to the total increase. On a regional scale, the extended drylands contribute around 13 % in Australia and 10 % in southern Africa to the increased dust emissions. Only in southern South America, more than 80 % of the increased dust emissions during the LGM came from the extended drylands (see Table 4). However, dust emissions also depend on meteorological factors like wind speed and soil moisture. Figure 2 shows anomalies (LGM - PI) with respect to the 2 m

temperature, annual precipitation and the 10 m wind speed. The ice sheet over North America and northern Europe caused a significant temperature drop in the according regions during the LGM (Fig. 2a), while the decrease around the equator was less pronounced, which resulted in a globally-averaged lower temperature of -4.1° C. The steeper temperature gradient between high latitudes and the equator during the LGM (Fig. 2b) caused noticeably stronger winds in the Northern Hemisphere (Fig. 2e, f), and the generally colder climate resulted in a precipitation decline at the equator and close to the poles (Fig. 2c, d), averaging to a global anomaly of -6.8 cm yr$^{-1}$. Since precipitation acts as an effective deposition mechanism, the drier climate north of 45° N also contributes to the higher dust burden towards the North Pole as a result of the increased particle lifetime, in addition to the higher dust emissions in Asia (Fig. 1f). In the Northern Hemisphere the wind speed anomaly averages to +0.49 m s$^{-1}$, and Fig. 2e reveals that there was an increase over the major dust source regions Asia and North Africa. Although the zonally-averaged wind speed in Fig. 2f suggests no significant differences between LGM and PI in the Southern Hemisphere (+0.04 m s$^{-1}$), the map shown in Fig. 2e indicates considerable differences over the major Southern Hemisphere dust source regions. A regional analysis yields a 10 m wind speed anomaly of +0.77 m s$^{-1}$ over Australia, +0.44 m s$^{-1}$ over South Africa and +0.52 m s$^{-1}$ over Patagonia. With respect to precipitation, the model simulates a decrease of -31.7 cm yr$^{-1}$ for Australia, -28.2 cm yr$^{-1}$ for South Africa and a slight increase of +7.1 cm yr$^{-1}$ for Patagonia (Fig. 2c). Those findings are in agreement with results by Rojas et al. (2009), who found in the scope of the PMIP2 simulations a generally colder, drier climate south of 40° S, although slightly more precipitation over Patagonia during the LGM. Finally, the dust emissions also depend on the vegetation cover. As can be seen in Fig. 2g and h, arid and semi-arid areas were substantially further extended during the LGM compared to PI, in particular in Asia, Australia, Southern Africa and Patagonia.

Based on our findings, we conclude that the increase in dust emissions during the LGM compared to PI was caused by extended drylands in the source regions and at the coasts due to a lower sea level, reduced vegetation cover as well as regionally increased wind speeds and less precipitation over the source regions, which is also in agreement with findings by Lunt and Valdes (2002).

### 3.2.2 Comparison to observational data

We use the compilation of dust deposition data from Kohfeld et al. (2013) for a comparison between our simulated dust deposition in the Southern Hemisphere and data based on marine sediment and ice core analysis.

Figure 3 shows dust deposition maps including observations and scatter plots comparing simulated and measured dust deposition for PI, LGM and the LGM-PI-ratio. For a more differentiated analysis, the data points have been categorized into five groups according to their geographical region. For both PI and the LGM, the simulated values are mostly well in accordance with the observed values for the Atlantic Ocean (Fig. 3b, d, blue diamonds). Due to the atmospheric circulation, dust deposited in the region close to the equator originates mainly from the Sahara Desert as well as southern Africa, whereas dust deposited in the southern Atlantic Ocean is mainly of southern South American origin. The model reveals a tendency to underestimate the depositions over the Pacific / Pacific Southern Ocean, in particular in the PI simulation (Fig.

3b, d, green circles and triangles), as well as a systematic overestimation of the dust depositions onto Antarctica (Fig. 3b, d, orange pentagons) by one order of magnitude. At least for the Pacific region, Australia can be assumed to be the major source, which indicates that the model simulates too low dust emissions. However, the dust deposited over the Tasman Sea is also very likely of Australian origin (Fig. 3b, d, red crosses). While the simulation values are in good accordance with the observations in that region for PI, the model overestimates the dust deposition during the LGM. Fig. 3e, f shows a comparison between the simulated dust deposition ratios LGM/PI and the measurement values, suggesting that this ratio varies strongly on a regional scale. For instance, our simulation results indicate a 5 to 40-fold increase in dust deposition over Antarctica, whereas a 1 to 4-fold increase has been simulated over the Atlantic (Fig. 3f).

As our analyses of dust provenance (section 3.2.3) reveal, the dust deposition over Antarctica is dominated by the Australian contribution during the LGM. Thus, the comparison of our model results with the observational data showing lower simulated values in the Pacific / Pacific Southern Ocean gives no clear hint whether Australia's source strength is over- or underestimated in the model. A possible factor contributing to this inconsistency could be a too high transport efficiency of dust towards the South Pole. However, considering the differences in absolute values between dust deposition over the Pacific and Antarctica, which is almost three orders of magnitude, a too high transport efficiency can be excluded as the sole reason for the discrepancies. Another factor that needs to be considered are non-aeolian contributions. Although data from "marine sites that have been flagged because they are located within zones of thick nepheloid layers and ice-rafted detritus, which can contaminate aeolian signals" had already been excluded from the dataset we use for comparison (Kohfeld et al., 2013), the reconstructed detrital flux estimates might still contain contributions from glacier erosion and riverine input, which are not considered in our model. The dust flux reconstructions are based on the assumption of relatively constant proportions of $^{232}$Th in continental lithogenic materials and might thus be overestimated by 30 – 40 % in regions receiving fine-grained dust from Patagonia and Australia since fine sediments have "a reduced proportion of low-$^{232}$Th phases such as quartz and feldspar" (McGee et al., 2015). The lack of non-aeolian contributions in our model might also contribute to the fact that the simulated dust deposition fluxes appear so stratified in the Pacific / Pacific SO region compared to the observational data (Fig. 3b, d), but could also indicate a model shortcoming in the representation of the dust deposition process on small scales. Finally, it should be taken into account that we compare (simulated) aeolian dust deposition fluxes onto the ocean surface to marine sediment data, i.e., also any horizontal transport processes of the sediments deposited in the ocean are not considered.

Sediment core analysis suggests that the total dust deposition flux during glacials over the Pacific Southern Ocean (SO) is about 50 % of that over the Atlantic (Lamy et al., 2014), which can also be observed in our simulation results (Fig. 3d), revealing that the average dust deposition flux is around 1 g m$^{-2}$ yr$^{-1}$ in the Pacific SO and around 2 g m$^{-2}$ yr$^{-1}$ in the Atlantic. The globally-averaged LGM-to-PI ratio of 5.6 found with ECHAM6.3-HAM2.3 (see Table 3) is slightly above the assumed 2- to 5-fold increase found in the literature (e.g., Kohfeld and Harrison, 2001). However, the 25-fold increase in dust deposition over Antarctica suggested by ice core data (Lambert et al., 2008) is in good accordance with our simulation results. Additionally, the observed 1 to 4-fold increase in dust depositions over the Atlantic is well captured by the model. As

a consequence of the underestimation in the PI simulation, the dust deposition ratio is systematically too high for the Tasman Sea and the Pacific Ocean (Fig. 3f). Though, the model yields acceptable results in terms of absolute values for LGM climate conditions.

### 3.2.3 Dust provenance studies

In order to identify the relative contribution of the major dust source regions southern South America, southern Africa and Australia to the total dust deposition in the Southern Hemisphere, four additional simulations for both PI and LGM were performed. In each of them, source regions in the Southern Hemisphere were independently switched on and off, while in all experiments all dust source regions in the Northern Hemisphere are still considered. The four simulations consist of a South America-only run (SAMonly, southern Africa and Australia switched off), a South Africa-only run (SAFonly, South America and Australia switched off), an Australia-only run (AUSonly, South America and southern Africa switched off) and a Northern Hemisphere-only run (NORTHonly, all sources in the Southern Hemisphere switched off) in order to identify the contribution of interhemispheric dust. Although New Zealand is discussed as a potential additional dust source during the LGM (Lamy et al., 2014; Koffman et al., 2021), our model only simulates dust emissions of less than 1 Gg yr$^{-1}$ from this region, which is effectively negligible compared to the simulated emissions of 748 Tg yr$^{-1}$ from Australia and 36 Tg yr$^{-1}$ from Patagonia, respectively. Since our model runs in the rather coarse spatial resolution T63 (horizontal grid size of approx. 1.8° x 1.8°), New Zealand's geographical expanse might only be marginally captured in our model and thus the source strength could be underestimated.

Fig. 4 reveals the relative contribution of the four major source regions South America (red), South Africa (green), Australia (blue) and the Northern Hemisphere (yellow) to the total dust deposition in the Southern Hemisphere. For both PI and LGM, the maps reveal the typical dust transport patterns in the Southern Hemisphere, which are the southeasterly trade winds for regions close to the equator and the westerly wind belt for regions in the high latitudes. Along the equator, dust deposition is dominated by interhemispheric dust originating from the Northern Hemisphere. For Antarctica, our experiments reveal for PI (Fig. 4a) that the deposited dust is coming from Australia and South America, of which the latter is predominating and contributes between 60 and 80 % of the dust deposition over West Antarctica. Despite a significant increase in dust source strength of South America (Fig. 1b, Table 3) during the LGM similar to Australia, its relative contribution to the total dust deposition over the Southern Ocean and Antarctica decreases (Fig. 4b). We find that dust deposited over the SO in the eastern half of the Southern Hemisphere originates mainly from southern South America, while dust deposited over the Pacific and Pacific SO is mainly of Australian origin. Generally, those two sources contribute in sum slightly more than 90 % to the total dust deposition over the SO and Antarctica (60° S – 90° S) for both PI and LGM conditions (Table 5).

Due to the interactive coupling of mineral dust in our model, we checked whether the reduced dust load in the Southern Hemisphere resulted in a shift in the zonally-averaged precipitation and found no shifts regardless of the switched-off sources. The procedure as well as the findings are in agreement with results by Evans et al. (2020), who performed

similar experiments in order to investigate the influence of the dust load asymmetry between the Northern and Southern Hemisphere on the location of the ITCZ as a consequence of the resulting asymmetric radiative forcing. They likewise found that the influence of the Southern Hemisphere is almost negligible compared to the Northern Hemisphere due to the much smaller contribution to the total global atmospheric dust load. Those findings raise the confidence that our experiments yield meaningful results. Our model results for PI agree well with those found by Li et al. (2008) for current climate conditions using the GFDL Atmospheric General Circulation Model AM2, in particular the dominance of southern South American dust deposited over West Antarctica as well as equal contributions of Australian and South American dust deposited over East Antarctica. The discussed combined contribution of Australia and South America of more than 90 % to the total dust deposition over the Southern Ocean and Antarctica is again in accordance with results by Li et al. (2008), who found a combined contribution of more than 85 %. However, our identification of Australia as the predominant source for dust deposited over Antarctica during the LGM is inconsistent with observational data from Antarctic ice cores. The characteristic ratios of Sr and Nd isotopes suggest southern South America to be the most likely dust source, possibly with minor contributions up to 15 % by Australian or South African dust sources (Basile et al., 1997; Delmonte et al., 2008). Our contradicting results clearly indicate a shortcoming on the modeling side and our further analysis is intended to fathom the mechanism causing this discrepancy.

Although both dust sources Australia and South America increase by a factor of around 15 in our model during the LGM compared to PI (Table 3), the absolute amount of dust coming from Australia clearly dominates the Southern Hemisphere. Only in a smaller region of East Antarctica, dust of South American origin contributes up to 40 % of the total deposition. The deposition pattern of Australian dust shown in Fig. 4b suggests an increased long-range transport, which can likely be attributed to the higher particle lifetime during the LGM as a consequence of the generally drier climate in the Southern Hemisphere (Fig. 2c, d). At first glance, it seems that this should apply to dust coming from both sources Australia and southern South America and might consequently not provide a possible explanation for the decrease in the relative contribution of southern South American dust to the total deposition. However, as discussed in section 3.2.1, we found that the regional climate over Patagonia as well as parts of the Atlantic SO turned out to be slightly wetter during the LGM (Fig. 2c). Since the westerly wind belt is responsible for the dust transport at these latitudes, this suggests that dust of South American origin was removed by scavenging with a higher efficiency, while dust coming from Australia might have had higher particle lifetimes due to the drier climate simulated over the Pacific / Pacific SO.

### 3.2.4 Particle lifetime and radius anomalies in the Southern Hemisphere during the LGM

In order to test this hypothesis of changed glacial particle lifetimes, we investigate the particle lifetime anomaly (Fig. 5a). The correlation between the particle lifetime anomaly and the precipitation anomaly (Fig. 5b) can easily be recognized, which is not surprising considering the fact that in section 3.1 wet scavenging was identified to be the predominant deposition mechanism in our model. Especially in regions close to the equator, for instance in the Pacific Ocean and the Indian Ocean, but also more southwards in the Atlantic Ocean / Southern Ocean around 60° S, the pattern of increased

precipitation causing shorter particle lifetimes can be clearly seen. To get more insights into the decrease of the relative importance of southern South American dust concerning the total dust deposition onto the Southern Ocean and Antarctica, the particle lifetime southwards of the horse latitudes (30° S) were analyzed, since dust closer to the equator is essentially transported north-eastwards by the trade winds. On average, the drier climate south of 30° S during the LGM leads to an increased particle lifetime of +0.52 days. However, using the results of our provenance studies, we find that the average particle lifetime of Australian dust increases by +1.12 days, whereas it decreases by -1.69 days for dust particles of southern South American origin. Since the particle lifetime eventually influences the transport range, the over-proportional importance of Australian dust in the Southern Hemisphere during the LGM compared to PI conditions (Fig. 4a, b), in particular southwards of 60° S, can thus be explained by changes in the regional precipitation patterns.

Combining our results concerning changes in particle lifetime with our provenance studies, we want to take up on the ongoing debate whether changes in source strength or atmospheric particle lifetime were mainly responsible for the increased dust concentration during the LGM found in Antarctic ice cores. Figure 5c and d as well as Table 6 show clearly that wet scavenging is the main deposition mechanism for mineral dust between 60° S and 90° S in our model, in particular over the Southern Ocean, which is in agreement with the study of Markle et al. (2018), who found that precipitation is the "principal barrier to aerosols reaching the poles". However, while the authors suggest that changes in particle lifetime are the main reason for the increased dust transport to Antarctica during glacials, the generally drier climate during the LGM simulated by our model leads only to a slightly higher particle lifetime on average. Of bigger importance in the scope of our simulations are the regional changes in precipitation, and thus in particle lifetime, because they eventually lead to Australia becoming the predominant source of dust deposited over Antarctica. Although our model overestimates the dust deposition over Antarctica for both PI and LGM by one order of magnitude (Fig. 3b, d), the simulated LGM-to-PI ratio of 14 (Tab. 5) is generally in good agreement with observations (Fig. 3e, f). Apparently, the almost 16-fold increase in dust source strength during the LGM compared to PI for both southern South America and Australia (Tab. 3) is necessary to achieve this accordance. Furthermore, the simulated increase in source activity during the LGM can be traced back to increases in wind speed over the source areas, reduced vegetation, a generally drier climate as well as extended source regions due to a lower sea level. In particular in southern South America, extended source regions contribute more than 80 % to the dust emissions during the LGM (Tab. 4). Our findings are in agreement with results by Wolff et al. (2010), who suggested that the variability in the non-sea-salt calcium flux (as an elemental marker for terrestrial dust most likely originating from South America) on glacial-interglacial timescales found in Antarctic ice cores were rather caused by changes in the source region than by changes in atmospheric particle lifetime.

Finally, we have a look at particle-size anomalies between the LGM and PI in the Southern Hemisphere. Hereby, the focus is only on the insoluble modes, since the model assumes perfectly internally mixed soluble modes and thus the according particles do not consist of mineral dust only. The simulations show a clear trend of increasingly coarser particles for both the accumulation as well as the coarse mode in the Southern Hemisphere between the source areas and the South Pole (Fig. 5e,

f). Additionally, our results suggest a correlation between coarser particles and increased particle lifetimes as well as reduced precipitation (Fig. 5a, b).

The observed size variability between cold and warm climate states has been used to draw conclusions about meteorological conditions. On the one hand, the analysis of mineral dust retrieved from marine sediments (Hovan et al., 1991) and Greenland ice cores (Steffensen, 1997) consistently showed the deposition of coarser dust particles during glacials, which has been interpreted either as a consequence of stronger winds or the result of decreased weathering (Mahowald et al., 2014). On the other hand, dust retrieved from Antarctic ice cores shows spatially varying and opposing trends with respect to particle sizes during cold climates. For instance, a particle-size analysis of mineral dust retrieved from the EPICA-Dome C (EDC) ice core (75°06'S, 123°21'E) indicates the deposition of finer particles during the LGM compared to deposition occurring during the warmer climate of the Holocene (10 kyr BP), while the same analysis for an ice core drilled at the Dome B (DB) location (77°05'S, 106°48'E) yields the deposition of coarser particles during the LGM compared to the Holocene (Delmonte et al., 2004). Since the mineralogical analysis of the dust particles shows clearly that for both locations the isotopic ratios match southern South American sources, it can be concluded that the deposited dust was of the same geographical provenance. Consequently, the observed differences are assumed to be caused by changes in the atmospheric circulation. As coarser particles tendentially have a shorter particle lifetime, those found at DB during the LGM are likely the result of shorter trajectories, while the finer particles as well as the increased sorting of the particles, expressed by a comparably small σ found for the LGM (Delmonte et al., 2002), indicate longer pathways from the source to the deposition area. The relative increase of finer particles in the EDC region during the LGM has also been confirmed by modeling studies (e.g., Mahowald et al. (2006)). The model used by Albani et al. (2012) also showed the deposition of slightly coarser particles in regions of East Antarctica and a shift to generally finer particles in most other regions of Antarctica during the LGM, as indicated by ice core data (Delmonte et al., 2002, 2004). The authors explain the regional variation in particle size on the one hand by reduced wet deposition during transport, which leads to the general shift to finer particles, and suggest on the other hand that size-selective dry deposition dominates in the interior of Antarctica, which in turn leads to the regional deposition of slightly coarser particles. Our model results support the suggested spatial variation in the respective predominant deposition mechanism (Fig. 5e, f). Similar opposing dust grain-size trends over glacial-interglacial timescales have also been found in marine sediments retrieved from the South Pacific and South Atlantic, respectively (van der Does et al., 2021).

Our findings of generally coarser particles for both the accumulation as well as the coarse mode in the Southern Hemisphere between the source areas and the South Pole without any remarkable regional differences are contradicting to the previous discussion. For technical reasons, the mass median radius and the standard deviation of the particle size distribution in the scope of the dust emission process are fixed parameters in our model for both the insoluble accumulation and the insoluble coarse mode (see Sect. 3.2.1 and Stier et al. (2005)). Consequently, the emission of coarser particles during the LGM due to stronger winds in the source regions must be excluded. Since the standard deviation of the particle size distributions are kept constant for all modes, HAM2.3 does also not account for features like a more efficient particle sorting

due to longer particle lifetimes as discussed above. The only mechanism in the model leading to increased particle sizes is the coagulation of insoluble dust particles with sulfuric acid particles of much smaller size (Vignati et al., 2004). Consequently, the observed particle-size anomalies in the Southern Hemisphere during the LGM compared to the PI shown in Fig. 5e and f can be attributed to an increased sulfur-coating of the insoluble dust particles as a consequence of the longer particle lifetimes (Fig. 5a) due to reduced precipitation (Fig. 5b). These findings were tested by running the same simulation while switching off all sulfate sources, and indeed the observed particle radius anomalies vanished.

### 3.2.5 Meridional dust transport in the Southern Hemisphere during PI and LGM

Although the particle-size distribution implemented in HAM2.3 does not allow for the investigation of size-dependent processes within a given mode during transport, general transport patterns on larger scales can still be studied. Hereby, the meridional transport from the source areas in the Southern Hemisphere towards the South Pole is of particular interest. As a consequence of the Coriolis force, southward-moving air parcels carrying mineral dust are generally deviated eastwards, leading predominantly to a zonal transport and distribution of dust by the Westerlies, whereas the meridional transport is caused by eddies (Li et al., 2010). Figure 6 shows the zonally-averaged dust mass concentration depending on the altitude south of 30° S for both the PI (Fig. 6a, b) and the LGM (Fig. 6c, d) as well as the mean dust transport height for all modes and times (Fig. 6e). The general meridional dust transport pathway can be understood as a result of prevailing convective cells. After their emission in the mid- to low-latitudes, dust particles are transported to the mid- and high troposphere (and potentially even higher to the tropopause, see below) along the polar front, i.e., the boundary between the Ferrel and the polar cell, by convection. While removal processes like sedimentation and dry deposition take place close to the ground, dust particles transported in the troposphere are mostly removed by wet scavenging. Dust particles transported at higher altitudes, however, remain significantly longer in the atmosphere and can only get removed once they reach lower altitudes as a consequence of the convergence and subsequent sinking of cold air masses (so-called subsidence) close to the South Pole (James, 1989). As can be clearly recognized in Fig. 6 for both the PI and the LGM, the finer particles of the accumulation mode are transported at higher altitudes compared to particles in the coarse mode. This effect is caused by the higher sedimentation velocities of larger particles and has also been found in other studies (Tegen and Fung, 1994). Delmonte et al. (2004) used this insight in order to explain the differences in dust grain size at DB and EDC during the LGM (see above) despite their geographical proximity and same dust provenance. The authors suggest that the measured difference in particle size and grading (i.e., low σ) are caused by finer particles being transported on longer trajectories in the upper atmosphere and deposit in regions of air subsidence, whereas coarse particles were transported to the respective deposition areas by comparably short trajectories in the troposphere. This opposing temporal trend with respect to the measured particle size of dust deposited at DB and EDC during the LGM-Holocene transition is proposed to be the result of vortex migration, i.e., the southwards movement of an area of preferential upper air subsidence on the according timescale (Delmonte et al., 2004).

While differences in transport height between particles of the accumulation and the coarse mode can be observed for a given time period, obvious differences also exist with respect to transport height between the PI and the LGM for a

given mode. During the PI, the majority of dust particles in the accumulation (Fig. 6a) and the coarse mode (Fig. 6b) seem to get transported quickly by convection from the source regions to a typical altitude of 11 km and 9 km, respectively, at around 50° S and then follow a rather meridional pathway southward mostly at the same altitude. The southwards-directed meridional transport of mineral dust during the LGM, however, does not exhibit such clear vertical and horizontal patterns. Instead, the altitude seems to increase continuously during the southward transport, reaching a maximum of around 8km and 6km for the accumulation (Fig. 6c) and the coarse mode (Fig. 6d), respectively. The obtained difference in mean transport height (Fig. 6e) between the PI and the LGM can be attributed to reduced vertical mixing and advection as a consequence of colder surfaces (Fig. 2a, b), leading to a higher dust concentration at lower levels (Albani et al., 2012). These results suggest that the polar cell was vertically less extended during the LGM compared to the warmer PI climate.

## 4. Conclusions

ECHAM6.3-HAM2.3 constitutes a state-of-the-art model providing an interactive coupling of mineral dust emissions to the atmospheric model depending on surface properties as well as meteorological factors. For present-day conditions, the model yields reasonable results for dust emission, burden and deposition close to the median of other studies performed in the scope of the global dust model intercomparison in AeroCom phase I. Generally, our model performs rather at the lower end of the 1000 to 4000 Tg yr$^{-1}$ of dust emission estimated by the IPCC, which is caused by the neglect of a supercoarse particle mode. The predominant representation of fine and coarse particles leads to slightly higher particle lifetimes, which in turn enables the long-range transport of mineral dust to remote regions and causes wet scavenging to become the most dominant deposition mechanism in our model.

For pre-industrial climate conditions, a comparison to other modeling studies as well as measurement data suggests that in absolute numbers, the simulated dust emissions and depositions are too low, in particular in the Southern Hemisphere. The discrepancy is greatest in the South Pacific, suggesting that the dust source strength of Australia is underestimated in the model. However, since the simulated dust deposition in the Tasman Sea as well as over Antarctica are in well agreement with, respectively, slightly higher than the observational data, this model-data mismatch cannot be easily explained by a sole model source strength deficit and non-aeolian contributions not considered in our model might play a crucial role. For LGM climate conditions, the simulated dust deposition fluxes agree well with measurement data. As a consequence of the underestimation of the dust cycle during PI, the according simulated globally-averaged LGM to PI ratio with respect to dust depositions of 5.6 is slightly above the 2 to 5 suggested based on measurement data. A regional analysis in the Southern Hemisphere shows that the increase in dust emissions of the major sources southern South America, southern Africa and Australia during the LGM can be attributed to a generally drier climate causing less precipitation (except over Patagonia), extended source regions due to a lower sea level and significantly stronger winds in the source regions combined with reduced vegetation.

Our dust provenance studies indicate that for both PI and LGM climate conditions, over 90 % of the dust deposited between 30° S and 90° S is either of Australian or South American origin. However, our model suggests that Australia

constituted the predominant source of dust deposited over Antarctica during the LGM. This result is inconsistent with several data studies suggesting based on isotope analysis that most of the dust deposited over Antarctica during the LGM is most likely of southern South American origin and clearly indicates a shortcoming on the modeling side. Albeit both sources South America and Australia show an almost equal increase in strength during the LGM compared to PI in our model, the relative contribution of South America decreases. This can be traced back to an average increase in particle lifetime of Australian dust during the LGM, whereas the average particle lifetime of South American dust decreases due to changes in regional precipitation pattern. Despite a slight increase in particle lifetime in the Southern Hemisphere during the LGM due to the generally drier climate, the almost 16-fold source strength increase of southern South America and Australia during the LGM compared to PI seems to be necessary in order to achieve a on average 14-fold increase in dust deposition over Antarctica in our simulations, which is in good accordance with observational data.

For both PI and LGM, the finer particles in the accumulation mode are transported at greater altitudes than coarse mode particles within the Southern Hemispheric troposphere due to lower sedimentation velocities. Additionally, both modes exhibit a clear trend of being transported at lower altitudes during the LGM, which can be explained by reduced convection due to colder surfaces and indicate that the Polar Cell was vertically less extended during the LGM.

Our study showed clearly the capabilities and limitations of ECHAM6.3-HAM2.3. In particular microphysical effects can only be studied to some degree since all particles within a given mode are assumed to have the same physical and chemical properties. Since SSTs influence precipitation patterns and other climate elements like wind speed, which in turn affect the dust emission process, prescribing different boundary conditions based on reconstructions suggesting cooler SSTs might turn out to be a useful approach to reduce the data-model discrepancy, in particular with regards to the provenance of dust deposited over Antarctica during the LGM. Future sensitivity studies might yield new insights on that matter.

## 5. Code availability

The ECHAM-HAMMOZ model code as well as all required input data are freely available after signing a license agreement on https://redmine.hammoz.ethz.ch/projects/hammoz.

## 6. Author contributions

SK performed the simulations, evaluated the results and wrote the paper with supervision by MW. GL and MvD helped shaping the final manuscript. All authors contributed expert knowledge in the interpretation and discussion of the results.

## 7. Competing interests

The authors declare that they have no conflict of interest.

## 8. Acknowledgements

Stephan Krätschmer acknowledges funding of this study in the scope of the AWI Strategy Fund project DustIron. The work was supported by the research topic "Ocean and Cryosphere under climate change" in the program "Changing Earth – Sustaining our future" of the Helmholtz Society. We thank the Center for Climate Systems Modeling (C2SM) at ETH Zurich for hosting and providing the ECHAM-HAMMOZ model code. A special thanks goes to Ina Tegen, Anne Kubin and Kerstin Schepanski from the Leibniz Institute for Tropospheric Research (TROPOS) for support with respect to the model choice in the scope of our project as well as valuable input concerning technical details of the ECHAM-HAMMOZ model. Computational resources were provided by the AWI computing centre. Additionally, we thank Paul Gierz for technical support. Finally, we want to thank our Editor Steven Phibbs at Climate of the Past as well as Eric Wolff and two anonymous referees for their helpful suggestions on our manuscript.

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

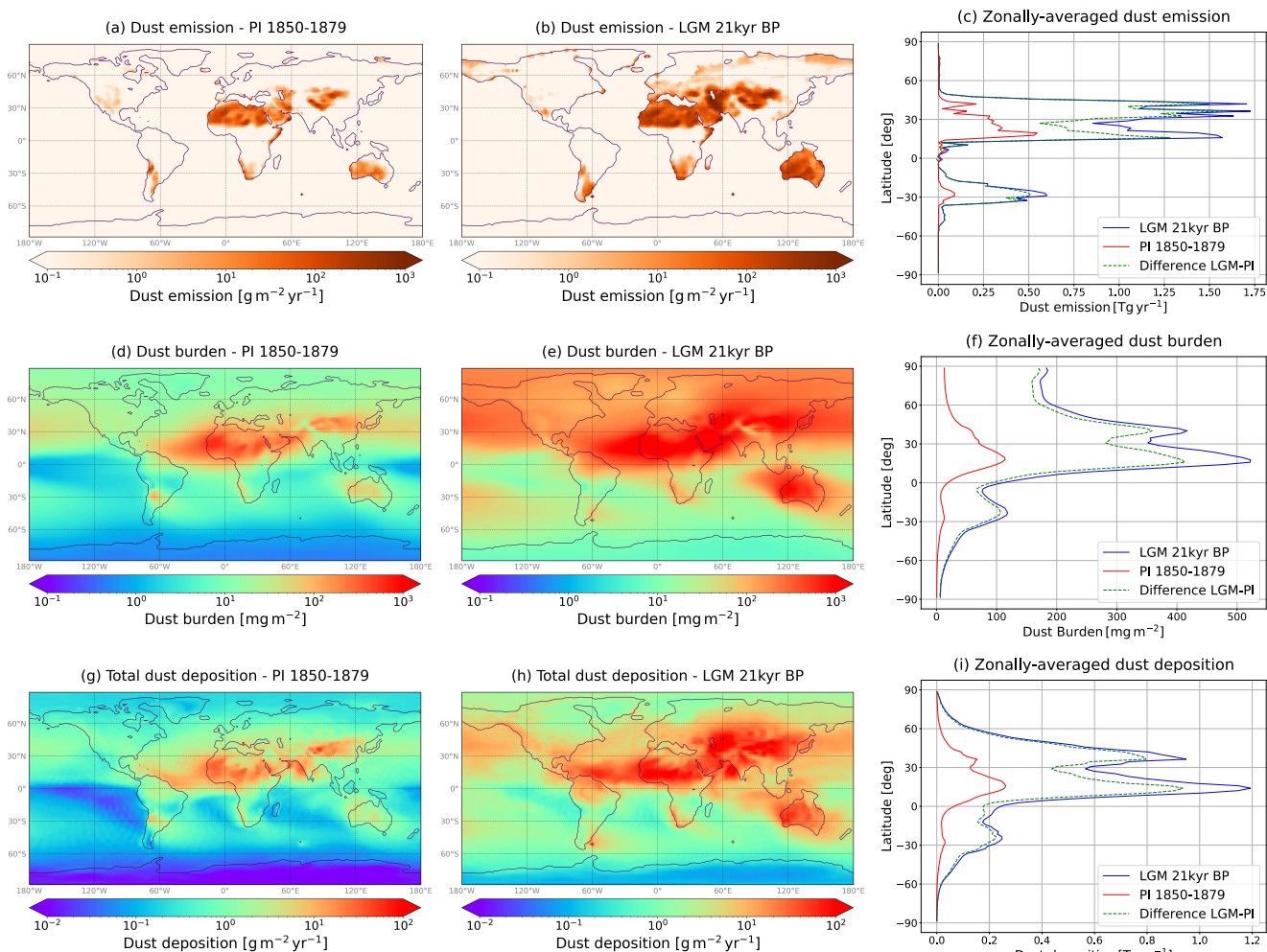

**Figure 1.** Global maps showing the dust emission, burden and deposition for PI (a, d, g) and LGM climate conditions (b, e, h). Additionally, zonally-averaged graphs are shown for all quantities (c, f, i).

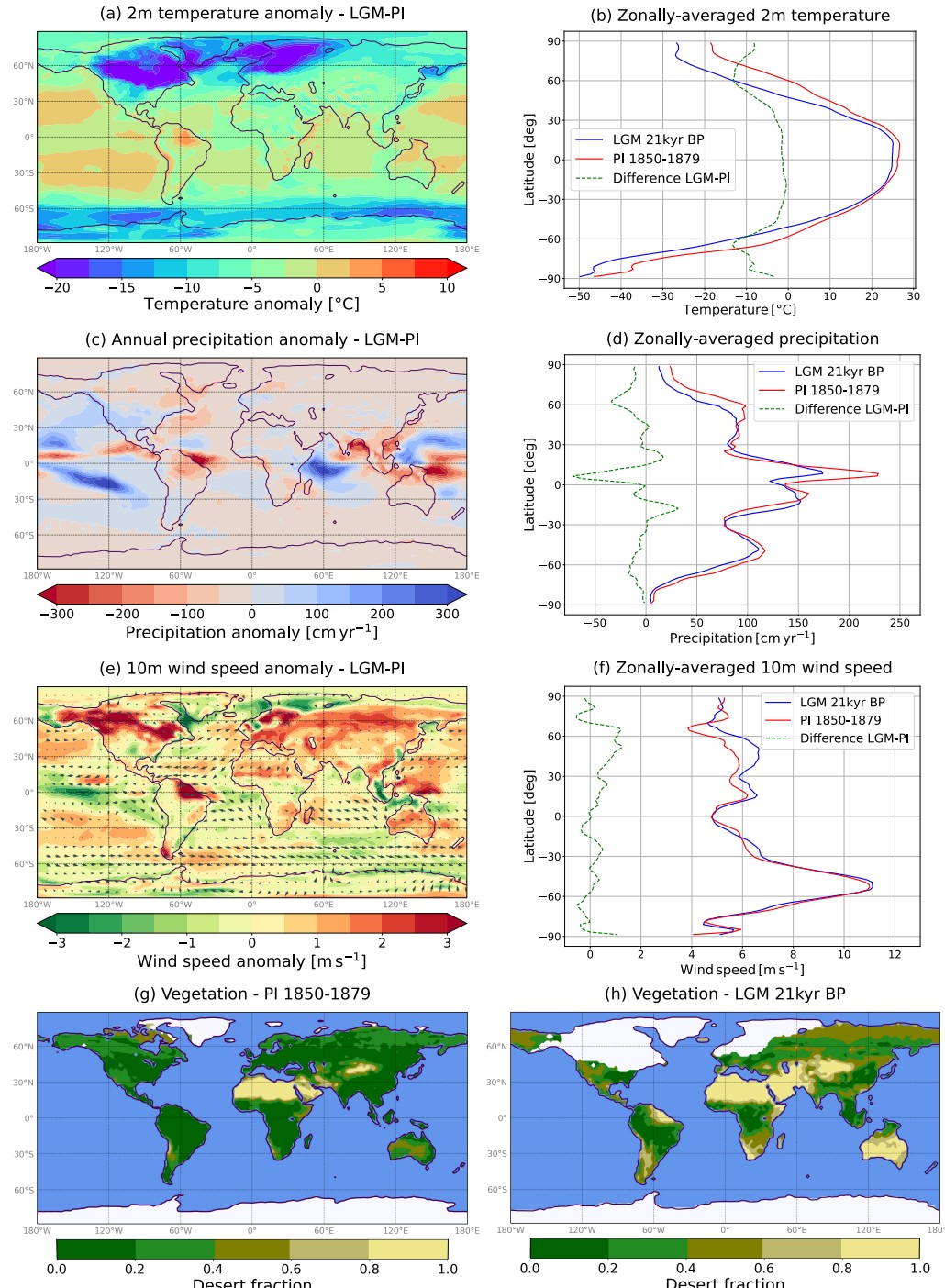

**Figure 2.** Global maps as well as zonally-averaged graphs of anomalies (LGM-PI) in the 2 m temperature (a, b), the annual precipitation (c, d), 10 m wind speed (e, f) and the desert fraction for each grid box for both PI and LGM (g, h). Please note that glaciers (white) and the ocean (blue) have been coloured in g and h for an improved readability.

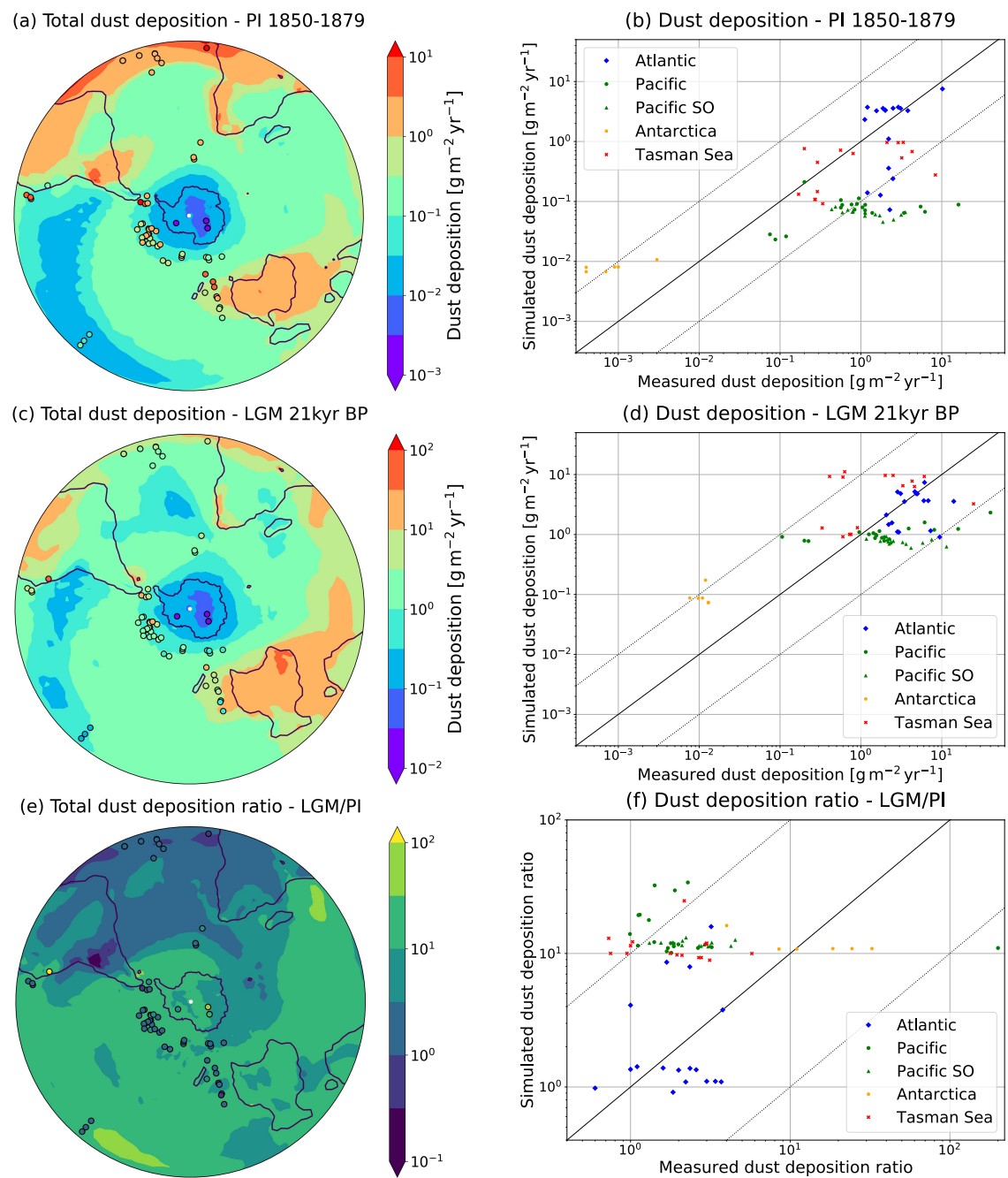


**Figure 3.** Comparisons between the simulated total dust deposition and observational data (Kohfeld et al., 2013) in g m⁻² yr⁻¹ from several regions in the Southern Hemisphere as well as the according scatter plots, for PI (a, b) and LGM (c, d). Figure (e) and (f) show the according simulated and measured dust deposition ratio LGM/PI. Please note the different value range in (a) and (c).

(a) Dust deposition contribution [%] - PI 1850-1879    (b) Dust deposition contribution [%] - LGM 21kyr BP

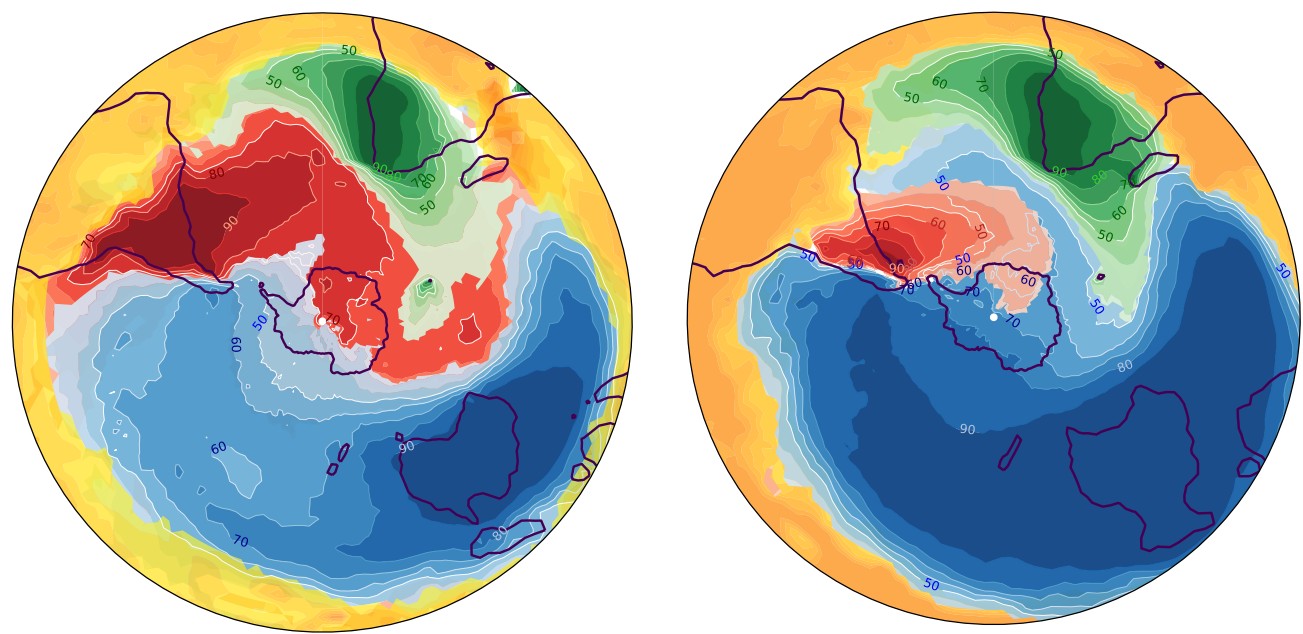


**Figure 4.** Results of the provenance studies showing the respective contribution of the major dust source regions South America (red), South Africa (green) Australia (blue) and the Northern Hemisphere (yellow) to the total dust deposition in the Southern Hemisphere in percent (white and blue numbers; different colours just chosen for improved readability) for PI (a) and LGM (b). Note that the Kerguelen Islands have been considered as a dust source in the SAFonly experiment and thus

appear in the same colour style applied for southern Africa.

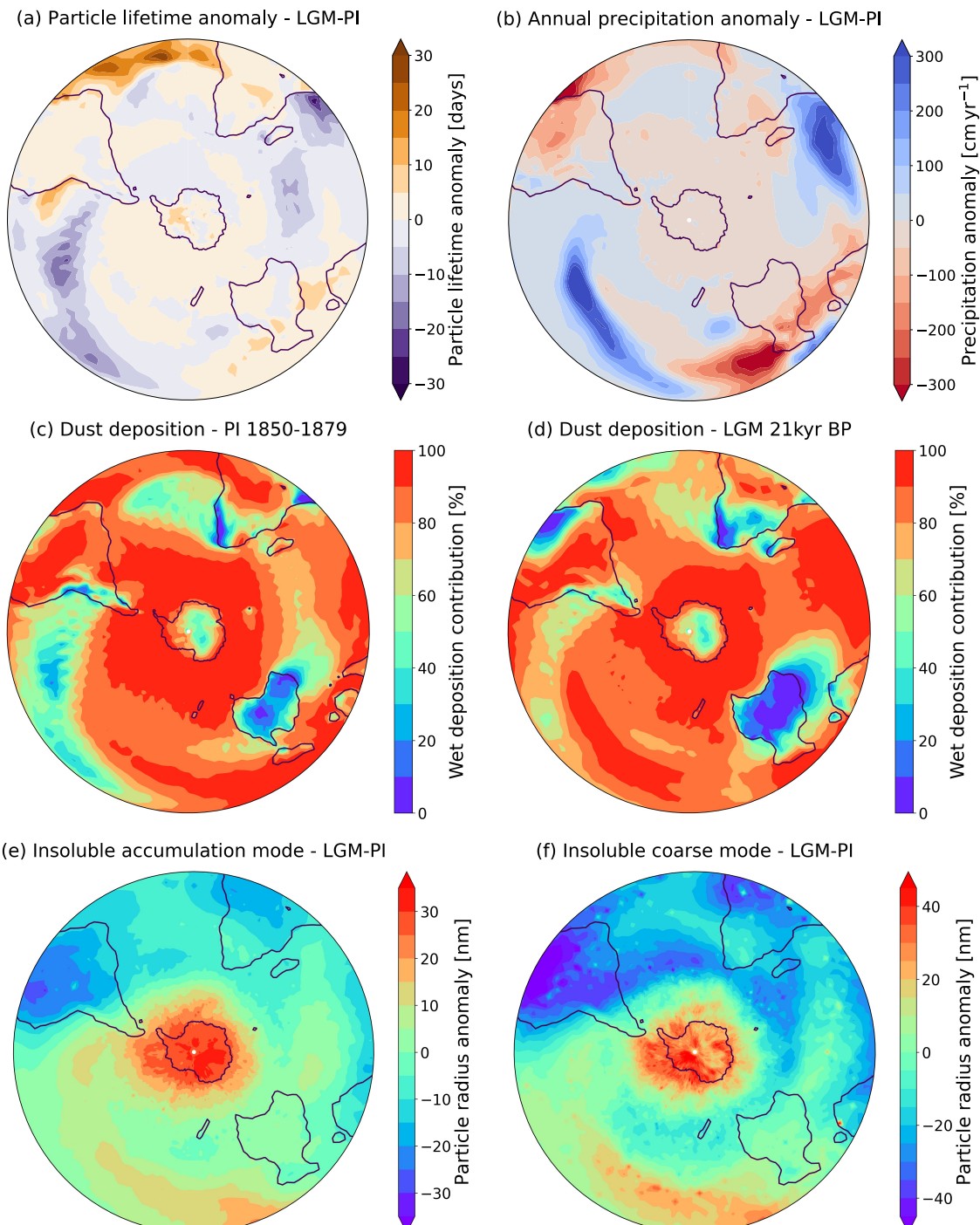

**Figure 5.** Particle lifetime anomalies (LGM-PI) in days (a), annual precipitation anomalies in cm yr$^{-1}$ (b) and the contribution of wet deposition to the total deposition in percent for both PI (c) and LGM (d) in the Southern Hemisphere. Additionally, particle radius anomalies in nm are shown for the insoluble accumulation (e) and insoluble coarse mode (f).

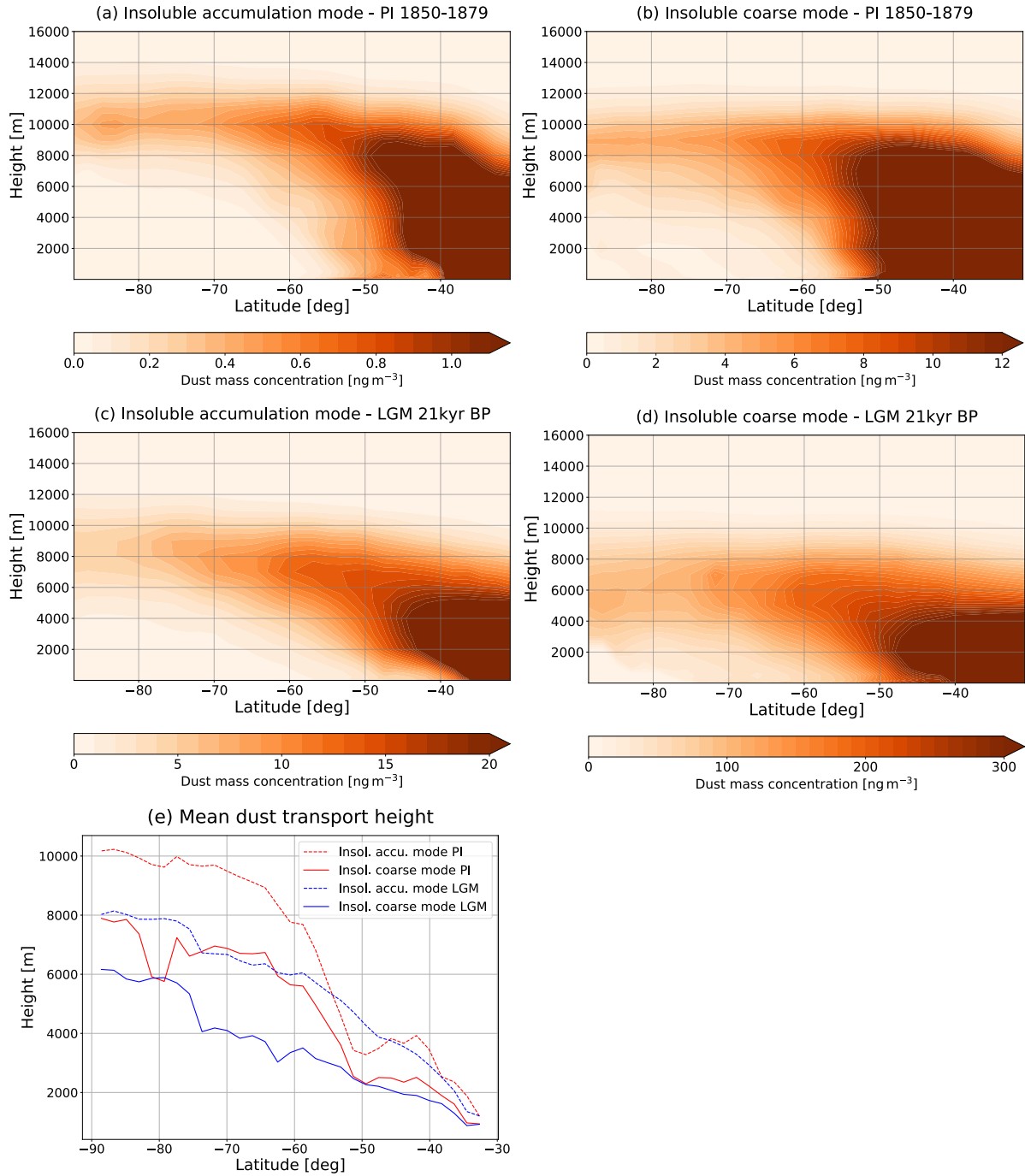


**Figure 6.** Zonally-averaged dust concentration depending on the altitude for the insoluble accumulation and coarse mode between 30° S and 90° S during PI (a, b) and LGM (c, d) as well as the mean transport height for all modes and simulation periods (e).

**Table 1.** Comparison of key values characterizing the global dust cycle budget between the model used in this study, ECHAM6.3-HAM2.3, and the AEROCOM median resulting from an intercomparison of 14 different atmosphere models for present-day climate conditions (Huneeus et al., 2011).

| Model / Experiment | Emission [Tg yr$^{-1}$] | Burden [Tg] | Deposition [Tg yr$^{-1}$] | Wet Deposition [Tg yr$^{-1}$] | Dry Deposition [Tg yr$^{-1}$] | Sedimentation [Tg yr$^{-1}$] |
|---|---|---|---|---|---|---|
| ECHAM6.3-HAM2.3 decadal mean 1996-2005 | 1221 | 19.8 | 1213 | 708 | 93 | 412 |
| AEROCOM median for year 2000 (range) | 1123 (514 - 4313) | 15.8 (6.8 – 29.5) | 1257 (676 - 4359) | 357 (295 - 1382) | 396 (37 - 2791) | 314 (22 - 2475) |

**Table 2.** Key values for dust emission, burden and deposition characterizing the global dust cycle budget as simulated by our model ECHAM6.3-HAM2.3 compared to results obtained with CESM (Albani and Mahowald, 2019).

| **Model** / Experiment | Emission [Tg yr$^{-1}$] | Burden [Tg] | Deposition [Tg yr$^{-1}$] | Wet deposition [Tg yr$^{-1}$] | Dry deposition [Tg yr$^{-1}$] | Sedimentation [Tg yr$^{-1}$] |
|---|---|---|---|---|---|---|
| **ECHAM6.3-HAM2.3** 30-year mean PI 1850-1879 | 923 | 16 | 929 | 547 | 57 | 325 |
| **CESM** PI year 1850 | 2785 | 20 | - | - | - | - |
| **ECHAM6.3-HAM2.3** 30-year mean LGM 21kyr BP | 5159 | 99 | 5171 | 3096 | 473 | 1602 |
| **CESM** LGM 21kyr BP | 6294 | 37 | - | - | - | - |

**Table 3.** Global and regional dust emissions during the PI and LGM in Tg yr$^{-1}$ as well as the LGM/PI ratio.

| Emission [Tg yr$^{-1}$] | PI 1850-1879 | LGM 21kyr BP | LGM/PI ratio |
|---|---|---|---|
| Globally | 923 | 5159 | 5.6 |
| Northern Hemisphere | 835 (90.6 %) | 4300 (83.3 %) | 5.3 |
| Southern Hemisphere | 88 (9.4 %) | 859 (16.7 %) | 9.8 |
| Saharan Desert | 535 | 1626 | 3 |
| Saudi Arabian Peninsula | 70 | 430 | 6.1 |
| Asia | 204 | 1803 | 8.8 |
| Australia | 47 | 748 | 15.9 |
| Southern Africa | 12 | 63 | 5.3 |
| Patagonia | 2.3 | 36 | 15.6 |

**Table 4.** Additional land areas globally as well as in Australia, Southern Africa and Patagonia during the LGM compared to PI due to the lower sea level in $10^6$ km$^2$ and the dust emissions from those extended drylands in Tg yr$^{-1}$.

| | Additional land area LGM [$10^6$ km$^2$] | Dust Emission [Tg yr$^{-1}$] PI 1850-1879 | Dust Emission [Tg yr$^{-1}$] LGM 21kyr BP | Dust Emission [Tg yr$^{-1}$] from additional land areas |
|---|---|---|---|---|
| Globally | 19.5 | 923 | 5159 | 229 |
| Australia | 1.8 | 47 | 748 | 92 |
| Southern Africa | 0.04 | 12 | 63 | 5 |
| Patagonia | 0.8 | 2.3 | 36 | 29 |

**Table 5.** Dust deposition onto Antarctica during the PI and LGM depending on the dust provenance in Tg yr$^{-1}$ and percent.

| Deposition region | | All sources [Tg yr$^{-1}$] | Australia [Tg yr$^{-1}$] | South America [Tg yr$^{-1}$] | South Africa [Tg yr$^{-1}$] | Northern Hemisphere [Tg yr$^{-1}$] |
|---|---|---|---|---|---|---|
| Southern Ocean | PI 1850-1879 | 1.04 | 0.38 (37 %) | 0.56 (54 %) | 0.09 (8 %) | 0.01 (1 %) |
| | LGM 21kyr BP | 13.48 | 9.25 (69 %) | 3.07 (23 %) | 0.96 (7 %) | 0.20 (1 %) |
| Antarctica | PI 1850-1879 | 0.21 | 0.06 (29 %) | 0.13 (62 %) | 0.02 (9 %) | 0.00 (0 %) |
| | LGM 21kyr BP | 2.88 | 1.96 (68 %) | 0.69 (24 %) | 0.19 (7 %) | 0.03 (1 %) |

**Table 6.** Contribution of dust deposition mechanisms to the total dust deposition between 30° S and 90° S during the PI and LGM in percent.

| Deposition region | | Wet deposition [%] | Dry deposition [%] | Sedimentation [%] |
|---|---|---|---|---|
| 30° S – 60° S | PI 1850-1879 | 79 | 8 | 13 |
| | LGM 21kyr BP | 68 | 13 | 19 |
| 60° S – 90° S | PI 1850-1879 | 91 | 1 | 8 |
| | LGM 21kyr BP | 92 | 0 | 8 |