# Peer review of "Simulating glacial dust changes in the Southern Hemisphere using ECHAM6.3-HAM2.3"

_Climate of the Past, 2021_

## Author Comment (AC1)

**Reply to Eric Wolff's comment on**

Krätschmer, S., van der Does, M., Lamy, F., Lohmann, G., Völker, C., and Werner, M.: *Simulating glacial dust changes in the Southern Hemisphere using ECHAM6.3-HAM2.3*, Clim. Past Discuss. [preprint], https://doi.org/10.5194/cp-2021-73, in review, 2021.

The paper by Krätschmer et al is a very welcome addition to the literature on dust modelling, particularly for the LGM. This adds to a number of studies such as those of Mahowald et al and it's excellent to have a new study using a more modern modelling set-up and with some novel diagnostics. It's an important topic because of its relevance to issues such as iron fertilisation, and its use in the interpretation of ice cores.

I do not intend to do a full review of the modelling aspects of the paper (best left to those with a modelling background), but rather to comment on particular issues related to what is seen in Antarctic ice cores. I enjoyed the description of what the model found and the comparison with previous modelling efforts. However I was rather astonished that the paper completely ignored recent discussions about the causes of increased dust in Antarctica during the LGM, and the extensive data papers that indicate a dominant South American source of dust across much of Antarctica in the LGM.

The latter issue (provenance) is the most glaring deficit in the paper. The authors conclude that Australia is the main source of dust to Antarctica in the LGM, with >70% contribution (Fig 4) over most of the continent. The authors then use this to discuss why other modelling studies might have got it wrong. In a very brief mention (line 401) it appears as if the authors are aware of the data (principally using Sr and Nd isotopes) showing a dominance of South American sources for the LGM (with a possibility of some Australian input in the Holocene, in contrast to their modelling results). This is not just a single result from one site, but is something documented at Vostok, Dome C, Talos Dome, and Taylor Glacier (e.g. Aarons et al 2017, Basile et al 1997, Delmonte et al 2008, Delmonte et al 2010). Given this obvious discrepancy between the modelling and the data it would surely be appropriate to either acknowledge that this is a discrepancy that implies an issue with the modelling, or offer reasons to suggest that the papers I mention have misinterpreted the geochemical data. It is certainly not OK to just ignore it, leaving the less informed reader with the misconception that it is likely that Australian sources dominate the Antarctic LGM dust budget.

We thank Eric Wolff for his very thoughtful and detailed comments on our manuscript.

It was never our intention to question the geochemical data regarding the provenance of dust found in Antarctic ice cores. Our model setup is certainly not suitable to question this data. The inconsistency highlights a problem on the modelling side and we will make this modelling deficit clearer in a revised manuscript.

The further discussion in the scope of our paper is intended to give insights on possible mechanisms causing Australia to be the predominant source of LGM dust deposited over Antarctica in our model simulations instead of southern South America, as indicated by the geochemical data. The model shows for both PI and LGM an overestimation of total dust deposition over Antarctica by a factor of 10 (*Fig. 3b, d*), of which 68% were contributed by Australia during the LGM (*Table 4*). Thus, this overestimation of total dust deposition in Antarctica could be caused by too much dust emitted from Australia in the model. However, the comparison of our model results to the marine data records of the Pacific SO region shows too little dust deposition in the simulations. These dust particles in the Pacific also stem from Australia (e.g. *Lamy et al., 2014*). These contradictory results for modeled dust depositions lead to the question whether the transport efficiency from Australia to Antarctica is overestimated. However, the absolute amounts of dust deposition over the Pacific SO and Antarctica differ by two orders of magnitude (*Fig. 3b, d*), i.e. even if less dust would get transported to Antarctica and deposit over the

Pacific SO instead, the discrepancy in this region would basically remain the same. The provenance studies for our model results allowed us to investigate changes in particle lifetime between PI and LGM for dust from each source individually. It turns out that the simulated wetter climate over southern South America and parts of the Atlantic SO (*Fig. 5b*) causes a decrease in particle lifetime (-1.69 days on average) of southern South American dust and an increase in particle lifetime (+1.12 days on average) of Australian dust (*Fig. 5a*) during the LGM. These changes in particle lifetime cause the deposition of a larger proportion of southern South American dust before transported to Antarctica, and longer transport ranges of Australian dust. Both effects combined lead to the Australian dust source dominance in our model.

Lamy, F., Gersonde, R., Winckler, G., Esper, O., Jaeschke, A., Kuhn, G., Ullermann, J., Martinez-Garcia, A., Lambert, F., and Kilian, R.: Increased Dust Deposition in the Pacific Southern Ocean During Glacial Periods, Science, 343, 403–407, https://doi.org/10.1126/science.1245424, 2014.

Less serious is that the paper ignores a quite strong debate in the ice core community about the relative importance of changes in source strength and lifetime in determining the LGM increase in dust concentration. Papers addressing this (sometimes discussing calcium as a dust proxy rather than dust per se) include (Wolff et al 2010, Fischer et al 2007, Petit et al 2009, Markle et al 2018). While the authors don't need to get into this debate in detail they could really offer some insight and it's a shame not to do so. The basic argument is that conceptual models suggest a big change in lifetime, while GCMs until now have not, having to rely on very big source changes to get the LGM dust increase. A question has been why the GCMs don't appear to document the change in dust lifetime one might expect due to the change in precipitation. The present paper is well-equipped to discuss this, mentioning that much of the transport is taking place above the level where precipitation occurs. However Fig 6 in the current paper suggests a new factor - that the transport level in the LGM is at lower altitude which does open the possibility of a second-order reason for a change in lifetime resulting from that (dust spending more time at altitudes where there is precipitation). Whether that change is really enough to explain the LGM increase (especially when in the current model the South American source sees a local precipitation increase) is not clear, but it would be valuable to see the discussion framed in this context.

Concerning changes in particle lifetime in our AGCM, we find the following results:

Generally, our model shows globally a particle lifetime increase (all modes) of +0.53 days during the LGM, and specifically +0.31 days in the Southern Hemisphere, which we attribute to the generally drier climate. However, *Fig. 5a* (*paper*) suggests opposing trends for the Pacific SO (increasing trend) and the Atlantic SO (decreasing trend) region. In the following, our discussion is based on values for burden and (wet) deposition (see *Table 1A*) for the two regions stated above for both PI and LGM. As shown in *Fig. 1A* and *Table 5* (*paper*), wet deposition is the predominant deposition mechanism (>90%) in the Southern Ocean region and consequently, we neglect contributions from dry deposition and sedimentation. *Table 1A* enables us to calculate the following particle lifetimes  $\tau$ :

 $\begin{aligned} \tau_{LGM,AtlanticSO} &\approx 6.8 \ days \\ \tau_{PI,AtlanticSO} &\approx 7.4 \ days \\ \tau_{LGM,PacificSO} &\approx 9 \ days \\ \tau_{PI,PacificSO} &\approx 7.4 \ days \end{aligned}$

The trends in particle lifetime changes clearly correlate with the precipitation anomaly plot shown in *Fig. 5b (paper)*. However, we want to emphasize at this point that *we do not state* that dust spends more time at altitudes where there is precipitation. For technical reasons, it is very difficult to determine the exact level at which precipitation occurs and is not possible based on our current simulation data. A more detailed investigation of microphysical processes could yield more insights on factors contributing to the obtained trends. The condensation of sulphate on particles in the insoluble modes leads to their transfer into soluble modes, where they grow in size due to water uptake until their sedimentation

---

## Author Comment (AC3)

**Reply to the 2[nd] anonymous referee's comment on**

Krätschmer, S., van der Does, M., Lamy, F., Lohmann, G., Völker, C., and Werner, M.: *Simulating glacial dust changes in the Southern Hemisphere using ECHAM6.3-HAM2.3*, Clim. Past Discuss. [preprint], https://doi.org/10.5194/cp-2021-73, in review, 2021.
* * *
**Review of Simulating glacial dust changes in the Southern Hemisphere using ECHAM6.3-HAM2.3 by Krätschmer, et al. for Climate of the Past.**

**The authors of this study use the ECHAM climate model to investigate the global variability of mineral dust emission, transport, and deposition across three different mean climates. (modern, pre-industrial, and LGM), with a particular focus on the Southern Hemisphere. The land model component includes dynamic vegetation that determines dust source locations, thereby permitting prognostic determination of changing dust source strength and location through time, and their LGM simulation includes enlarged coastlines to include exposed continental shelf among the potential dust sources. For each of their simulations the authors compare their results to observations and other model results, sometimes finding agreement and sometimes not. Their simulation of the LGM, and investigation of how it differs from the PI, is noteworthy as there are limited model studies of this time period that include dynamic dust sources. Of particular note are a series of results regarding the spatial variability of the provenance of dust deposited on Antarctica and the Southern Ocean. These results are a valuable contribution to the ongoing discussion regarding the relative contributions of South American and Australian sources through time, and thus the appropriate interpretation of archives of paleodust from the region. As this is a topic with some disagreement, the addition of results from a new model is welcome.**

**Overall I found the paper very well written and organized, and enjoyable to read. I do have some unanswered questions as well as some minor suggestions for the authors, so my recommendation is acceptance upon minor revisions.**

> Dear referee,
>
> thank you very much for reviewing our manuscript and the helpful suggestions you provided. Please find below our replies to your comments.

**Primary comments**

1. **The paper would benefit from more time in the introduction and conclusion spent reviewing what is known and the disagreements regarding dust provenance to Antarctic and the Southern Ocean. I was pleased when the authors brought up many of the studies when discussing their results, but I felt the bigger picture was somewhat overlooked. Specifically, there are conflicting studies regarding whether South America or Australia is the primary source of dust (and for that matter how much is contributed by Antarctic sources), and the relative role of dust source strength and transport efficiency. I think the authors have room here to set up and then answer some questions about how these discrepancies can be resolved by considering the time-varying relative strength of sources, the transport efficiency, and the spatial distribution of their influence. I think much of this information is already contained in the paper, but an explicit consideration of the debate would be valuable. One additional source to consider is Markle, et al. (2018).**

> Since the provenance studies for dust in the Southern Hemisphere are an important part of our study, we agree and will include a brief overview on the conflicting studies regarding whether South America or Australia is the primary source of dust already into our introduction section. As written in our reply to Eric Wolff's comment on our manuscript [1], we will not be able to give
* * *
[1] https://cp.copernicus.org/preprints/cp-2021-73/cp-2021-73-AC1-supplement.pdf

an ultimate answer on the question whether changes in source strength or transport efficiency eventually led to the observed increase in mineral dust aerosol concentration during the LGM found in marine sediments and Antarctic ice cores. Considering the fact that more than 90% of the dust deposition in the Southern Ocean region occurs due to precipitation in our model, we agree with Markle et al. (2018) that "precipitation in the mid-latitudes is the principal barrier to aerosol reaching the poles". However, we disagree that changes in the hydrologic cycle are the primary driver since we also find substantial and required changes in source strength by a factor of 16 for both Patagonia and Australia, which we can trace back well to changes in vegetation, wind speed, soil moisture and extension of the source areas. Moreover, the authors find the best agreement between their modeling results and data "at multi-centennial and longer timescales", while our study captures only a 30-year period under LGM conditions and thus considers in particular processes on much shorter timescales. We will include those aspects in our revised manuscript.

2. **What about New Zealand? I was surprised that the LGM simulations don't seem to include an expanded dust source from the exposed continental shelf around New Zealand, nor any discussion of it as a dust source during that period. Neff, et al. (2015) and Koffman, et al. (2021) would be relevant to this discussion.**

We are aware of the ongoing discussions on New Zealand as a potential dust source especially for the South Pacific during the last LGM (e.g. Lamy et al., 2014). Our model yields annual dust emissions of less than 1 Gg yr$^{-1}$ from New Zealand during the LGM, which is effectively negligible compared to the simulated emissions of 748 Tg yr$^{-1}$ (Australia) and 36 Tg yr$^{-1}$ (Patagonia). The low emissions in our model also explain why New Zealand appears to not represent a dust source at all during the LGM in Fig. 1b (*paper*).

One reason to consider is that New Zealand's geographical expanse is rather small and thus only marginally captured by our model running in the spatial resolution of T63, which corresponds to a grid box size of approximately 180 km (Sidorenko et al., 2015). Consequently, New Zealand's source strength and the according contribution to the dust deposition in the Pacific sector of the Southern Ocean during the LGM (Koffman et al., 2021) might be underestimated in our model. The study of Neff and Bertler (2015) uses a trajectory modeling approach for the years 1979 to 2013 based on reanalysis datasets for the according pressure fields. Additionally, the authors mention in their study that they investigate solely the trajectories without taking into account emission and deposition processes, which does not enable us to compare our simulation results for the LGM to the results of their study.

However, we agree that our findings concerning New Zealand's role as a dust source during the LGM should be included in our study and we will do so in our revised manuscript.

3. **The authors cover many of the modeling studies of dust transport to the Antarctic in their discussion of provenance, but there are also studies that take an isotopic approach that should be discussed as well. Wengler et al. (2019), McGee et al. (2016).**

The study of Wengler et al. (2019) provides lithogenic flux data only for the Holocene, stating that surface sediments near New Zealand "most likely indicate a combination of Australian dust and riverine input from New Zealand". Since our model only considers aeolian dust fluxes, we cannot compare our simulation results for PI and LGM.

However, we will include in our revised manuscript that the reconstructed dust fluxes used in our study for comparison with our simulation results (DIRTMAP, Kohfeld et al. (2013)), which are based on the assumption of relatively constant proportions of $^{232}$Th in continental lithogenic materials, might be overestimated by 30–40 % in regions receiving fine-grained dust from Patagonia and Australia (McGee et al., 2015).

**Minor comments**

1. **How is land tiling / vegetation coverage determined for the newly exposed continental shelf? Essentially, is the newly exposed land always a dust source, or can it become vegetated?**

   As mentioned in our manuscript, we perform a restart for our LGM experiments using restart files, which represent a dynamic equilibrium of our model for the according topographic, vegetation and climate conditions obtained after several hundred years of simulation (line 172).

   Generally, the land fraction of *each* grid cell can become vegetated according to the rules and equations for dynamic vegetation implemented in the land surface and vegetation model JSBACH, which are described in detail in Reick et al. (2013), paragraph 3 "*Natural Land Cover Change*". In short, the process of increasing / decreasing vegetation for each grid cell depends on several factors like meteorological conditions (temperature, precipitation, humidity, …), competition among the different plant functional types (PFT), the respective time constants for growth and lifetime for each PFT etc.. One of the leading principals is the so-called *universal presence*, i.e. seeds of each PFT are always on the land fraction in each grid cell available and the factors mentioned before determine the respective proportion of each PFT per grid cell. The land fraction of each grid cell can be further divided into a vegetated and a non-vegetated (desert) fraction. The latter has been used by (Stanelle et al., 2014) to determine the area for dust emissions in each grid cell, which are additionally influenced by several meteorological factors like wind speed, soil moisture etc.. The following maps show the desert fraction for each grid cell for PI and LGM.

[Figure]

2. **Since additional land area during the LGM is credited as one of the causes of increased dust emission, I would like some information, similar to the reported wind and precip changes, that tells me how much additional land there is in each region (and possibly some discussion of how much of the dust is being created from this new land).**

   Please find the according information in the following table.

| | Additional land area LGM [Mio. km²] | Dust Emission [Tg yr⁻¹] PI 1850-1879 | Dust Emission [Tg yr⁻¹] LGM 21kyr BP | Dust Emission [Tg yr⁻¹] on additional land areas |
|---|---|---|---|---|
| **Globally** | 19.5 | 923 | 5159 | 230 |
| **Australia** | 1.8 | 47 | 748 | 92 |
| **Southern Africa** | 0.04 | 12 | 63 | 5 |
| **Patagonia** | 0.8 | 2.3 | 36 | 29 |

   Globally as well as in Australia and Southern Africa, the additional land areas as a consequence of the lower sea level during the LGM contribute only a small proportion to the additionally emitted dust. Consequently, the main reasons for the increase are changes in vegetation (see maps above) and meteorological factors, for instance precipitation patterns and wind speed. In Patagonia, however, the additional land area, mainly consisting of drylands (see vegetation

maps), contributes substantially to the absolute dust emissions, which can also be seen in Fig. 1b (*paper*).

3. **In Figure 3, why are the simulated dust deposition values so stratified? The observations appear continuous across a couple orders of magnitude, while the simulated deposition values appear to form horizontal lines.**

   The stratification of the simulated dust deposition values indicate that the model is not able to capture the observed variation for both PI and LGM, in particular in the Pacific / Pacific SO region. Since the measurement locations are rather close to each other, the discrepancy might be caused by a shortcoming in the representation of the deposition process on a small scale in our model. However, it should be taken into account that we compare aeolian dust deposition fluxes onto the ocean surface to marine sediment data, i.e. aside from potential shortcomings in our model, horizontal dust transport processes in the ocean during sedimentation as well as dust flux contributions due to glacier erosion and fluvial inputs are not considered and might play a crucial role (e.g. Trudgill et al. (2020))

4. **Continuous colorbars on log plots are difficult to accurately interpret. The maps in Figures 3 and 1 would be much easier to read if the colorbars had discreet steps (while keeping the log scale). In Figure 1 it didn't bother me to much because I was more interested in the qualitative pattern than the quantitative values, but Fig. 3e I wanted o know where the one contour was, which is quite difficult to tell. I would suggest colorbars similar to those in Figs 2 and 5.**

   Thank you for this suggestion, please find the accordingly adjusted plots (Fig. 3a, c, e) below.

[Figure]

**References**

Markle, et al. (2018) Concomittant variability in high-latitude aerosols, water isotopes, and the hydrologic cycle, Nature Geoscience.

Neff, et al. (2015) Trajectory modeling of modern dust transport to the Southern Ocean and Antarctica, JGR: Atmospheres.

Koffman, et al. (2021) New Zealand as a source of mineral dust to the atmosphere and ocean, Quaternary Science Reviews.

McGee, et al. (2016) Tracking eolian dust with helium and thorium: impacts of grain size and provenance, Geochimica et Cosmochimica Acta

Wengler, et al. (2019) A geochemical approach to reconstruct modern dust fluxes and sources to the South Pacific, Geochimica et Cosmochimica Acta

References

Koffman, B. G., Goldstein, S. L., Winckler, G., Borunda, A., Kaplan, M. R., Bolge, L., Cai, Y., Recasens, C., Koffman, T. N. B., and Vallelonga, P.: New Zealand as a source of mineral dust to the atmosphere and ocean, Quat. Sci. Rev., 251, 106659, https://doi.org/10.1016/j.quascirev.2020.106659, 2021.

Kohfeld, K. E., Graham, R. M., de Boer, A. M., Sime, L. C., Wolff, E. W., Le Quéré, C., and Bopp, L.: Southern Hemisphere westerly wind changes during the Last Glacial Maximum: paleo-data synthesis, Quat. Sci. Rev., 68, 76–95, https://doi.org/10.1016/j.quascirev.2013.01.017, 2013.

Lamy, F., Gersonde, R., Winckler, G., Esper, O., Jaeschke, A., Kuhn, G., Ullermann, J., Martinez-Garcia, A., Lambert, F., and Kilian, R.: Increased Dust Deposition in the Pacific Southern Ocean During Glacial Periods, Science, 343, 403–407, https://doi.org/10.1126/science.1245424, 2014.

Markle, B. R., Steig, E. J., Roe, G. H., Winckler, G., and McConnell, J. R.: Concomitant variability in high-latitude aerosols, water isotopes and the hydrologic cycle, Nat. Geosci., 11, 853–859, https://doi.org/10.1038/s41561-018-0210-9, 2018.

McGee, D., Winckler, G., Borunda, A., Serno, S., Anderson, R., Recasens, C., Bory, A., Gaiero, D., Jaccard, S., Kaplan, M., McManus, J., Revel, M., and Sun, Y.: Tracking eolian dust with helium and thorium: Impacts of grain size and provenance, Geochim. Cosmochim. Acta, 175, https://doi.org/10.1016/j.gca.2015.11.023, 2015.

Neff, P. D. and Bertler, N. A. N.: Trajectory modeling of modern dust transport to the Southern Ocean and Antarctica, J. Geophys. Res. Atmospheres, 120, 9303–9322, https://doi.org/10.1002/2015JD023304, 2015.

Reick, C. H., Raddatz, T., Brovkin, V., and Gayler, V.: Representation of natural and anthropogenic land cover change in MPI-ESM: Land Cover in MPI-ESM, J. Adv. Model. Earth Syst., 5, 459–482, https://doi.org/10.1002/jame.20022, 2013.

Sidorenko, D., Rackow, T., Jung, T., Semmler, T., Barbi, D., Danilov, S., Dethloff, K., Dorn, W., Fieg, K., Goessling, H. F., Handorf, D., Harig, S., Hiller, W., Juricke, S., Losch, M., Schröter, J., Sein, D. V., and Wang, Q.: Towards multi-resolution global climate modeling with ECHAM6–FESOM. Part I: model formulation and mean climate, Clim. Dyn., 44, 757–780, https://doi.org/10.1007/s00382-014-2290-6, 2015.

Stanelle, T., Bey, I., Raddatz, T., Reick, C., and Tegen, I.: Anthropogenically induced changes in twentieth century mineral dust burden and the associated impact on radiative forcing, J. Geophys. Res. Atmospheres, 119, 13,526-13,546, https://doi.org/10.1002/2014JD022062, 2014.

Trudgill, M. D., Shuttleworth, R., Bostock, H. C., Burke, A., Cooper, M. J., Greenop, R., and Foster, G. L.: The Flux and Provenance of Dust Delivered to the SW Pacific During the Last Glacial Maximum, Paleoceanogr. Paleoclimatology, 35, e2020PA003869, https://doi.org/10.1029/2020PA003869, 2020.

Wengler, M., Lamy, F., Struve, T., Borunda, A., Böning, P., Geibert, W., Kuhn, G., Pahnke, K., Roberts, J., Tiedemann, R., and Winckler, G.: A geochemical approach to reconstruct modern dust fluxes and sources to the South Pacific, Geochim. Cosmochim. Acta, 264, https://doi.org/10.1016/j.gca.2019.08.024, 2019.

---

## Author Response (AR1)

**Detailed replies to both anonymous referees comments on**

Krätschmer, S., van der Does, M., Lamy, F., Lohmann, G., Völker, C., and Werner, M.: *Simulating glacial dust changes in the Southern Hemisphere using ECHAM6.3-HAM2.3*, Clim. Past Discuss. [preprint], https://doi.org/10.5194/cp-2021-73, in review, 2021.

**Referee 1:**

| Comment by referee 1                                                                                                                                                                                                                                                                                                                | Reply by authors                                                                                                                                                                                                                                                                                                                                                                                                                                                                                                                                                                                                                                                                                                                                                                                                                                                                                                                                                                                                                                                                                                                             | Changes in the manuscript                                                                                       |  |
|-------------------------------------------------------------------------------------------------------------------------------------------------------------------------------------------------------------------------------------------------------------------------------------------------------------------------------------|----------------------------------------------------------------------------------------------------------------------------------------------------------------------------------------------------------------------------------------------------------------------------------------------------------------------------------------------------------------------------------------------------------------------------------------------------------------------------------------------------------------------------------------------------------------------------------------------------------------------------------------------------------------------------------------------------------------------------------------------------------------------------------------------------------------------------------------------------------------------------------------------------------------------------------------------------------------------------------------------------------------------------------------------------------------------------------------------------------------------------------------------|-----------------------------------------------------------------------------------------------------------------|--|
| 30-33: More precisely, dust scatters
and absorbs both SW and LW
radiation, although scattering prevails
in SW (still, the single scattering
albedo of dust is not equal to 1, e.g.
Balkanski et al., 2007) and absorption
in the LW (although scattering may be
important too, e.g. Dufresne et al.,
2002). | We will rephrase the sentence as follows:
"During transport, the dust particles directly
influence Earth's radiation budget by scattering
and absorbing short- and longwave radiation
depending on particle size and mineralogical
composition (Dufresne et al., 2002; Balkanski et
al., 2007), which in turn affects the atmospheric
stability by altering the vertical temperature
profile and relative humidity (Boucher, 2015)."                                                                                                                                                                                                                                                                                                                                                                                                                                                                                                                                                                                                                                                                                 | We rephrased the sentence accordingly in
the lines 30–33.                                                    |  |
| 63-65: "We compare present-day
simulation results to model results".
Please rephrase.                                                                                                                                                                                                                                         | We will rephrase the sentence as follows:
"We compare our present-day simulations to
results obtained in the scope of the global dust
model intercomparison in AeroCom phase I in
order to []"                                                                                                                                                                                                                                                                                                                                                                                                                                                                                                                                                                                                                                                                                                                                                                                                                                                                                                                                   | We rephrased the sentence accordingly in the lines 82-84.                                                       |  |
| 165-169: It's not very clear to me what
these regional correction factors are
exactly, and how they are applied to
the present study, to maximize the
match with which observations and
how, or what are they values. Please
clarify the procedure in more detail.                                                | The regional correction factors are a natural consequence of the parameterization of a sub-
grid process on a mm scale in a model running with a typical resolution of 100 km and is essentially a mean to compensate for the lack of required information for an exact calculation of the considered process. A precise explanation can be found in Tegen et al. (2019):
"In previous versions, a global correction factor of 0.86 was applied on the threshold friction velocity to account for the inhomogeneity of the factors influencing dust emissions (e.g., surface wind) across the rather coarse model grid boxes. In ECHAM6.3 the surface orography is not taken into account for the aerodynamic surface roughness, in contrast to earlier versions. The subsequent changes in surface wind distributions over dust source areas require additional regional correction factors. For each relevant region that contains dust sources the correction factors are chosen such that the emissions agree with the values by Huneeus et al. (2011). These regional correction factors can be modified via the model namelist." | We included the according information as
well as a reference to Tegen et al. (2019) in
the lines 185-188. |  |
| 176: "Since our simulation periods are
comparably short" compared to
what? I do not understand this
passage. I gather you use an
atmosphere only model coupled to
land surface scheme and consider
prescribed SST for the ocean surface.
Okay, so how does this sentence fit
into that? Please rephrase.    | This sentence has been used to emphasize that
the temporal change in the interaction (more
precisely, the heat exchange) between ocean and
atmosphere can be neglected due to the much
higher inertia of the ocean surface, i.e. $\tau_{oc} \gg$
$\tau_{At}$ . Since taking into account the spatio-
temporal development of SIC and SST would
imply coupling a complete ocean/sea-ice model
to our current setup, this approximation saves a                                                                                                                                                                                                                                                                                                                                                                                                                                                                                                                                                                                                                                                                        | We rephrased the sentence accordingly in the lines 198-199.                                                     |  |

|                                                                                                                                                                                                                                                                                                                                                                                                                                                                                                                                                                                                                                                                                                                                                                                                                                                                            | significant amount of computational resources.
Instead, a constant external forcing file of SSTs
representative for the considered time period is
prescribed.                                                                                                                                                                                                                                                                                                                                                                                                                                                                                                                                                                                                                                                                                                                                                                                                                                                                                                                                                                                                                                                                                                                                                                                                                                                                                                                                                                                                                                                                           |                                                                          |
|----------------------------------------------------------------------------------------------------------------------------------------------------------------------------------------------------------------------------------------------------------------------------------------------------------------------------------------------------------------------------------------------------------------------------------------------------------------------------------------------------------------------------------------------------------------------------------------------------------------------------------------------------------------------------------------------------------------------------------------------------------------------------------------------------------------------------------------------------------------------------|--------------------------------------------------------------------------------------------------------------------------------------------------------------------------------------------------------------------------------------------------------------------------------------------------------------------------------------------------------------------------------------------------------------------------------------------------------------------------------------------------------------------------------------------------------------------------------------------------------------------------------------------------------------------------------------------------------------------------------------------------------------------------------------------------------------------------------------------------------------------------------------------------------------------------------------------------------------------------------------------------------------------------------------------------------------------------------------------------------------------------------------------------------------------------------------------------------------------------------------------------------------------------------------------------------------------------------------------------------------------------------------------------------------------------------------------------------------------------------------------------------------------------------------------------------------------------------------------------------------------------------------------------|--------------------------------------------------------------------------|
| 213-214: This statement is essentially
based on a set of global metrics
compared to Huneeus et al. (2011). It
is true that the dust scheme is
described in more detail Stanelle et al.
2014, and there validated against a
wide set of observations of other
features of interest for the
representation of the dust cycle;
however I would expect to see some
comparison here too, with the current
version of the ECHAM model setup,
also because it appears that some
tuning was done, and I found no
reference to another paper describing
it. The spatial patterns of dust
emissions indeed appear to show
some difference with Stanelle et al.
2014, also concerning the Southern
Hemisphere. Please add some more
information in this respect or an
appropriate reference if that exists
already. | Since it turned out that the model version already
came with a set of tuning factors matching the
results found in Huneeus et al. (2011) for present-
day conditions, we did no further tuning. The
tuning factors were only changed in the scope of
the provenance studies. Stanelle et al. (2014) only
shows the emission flux for present-day (PD) and
the anomaly PD-historic, while we show the
emission flux for pre-industrial (historic)
conditions and the LGM, which makes a direct
comparison rather difficult. However, comparing
the PD plot of Stanelle et al. (2014) to our PI plot,
we can still recognize the typical dust source
areas and emission patterns, in particular in the
Southern Hemisphere.
Concerning changes in absolute values for dust
emission etc., it needs to be emphasized that
Stanelle et al. (2014) used the model version
ECHAM6.1.0-HAM2.1-MOZ0.8 for their
simulations, while we used ECHAM6.3.02-
HAM2.3-MOZ1.0 for our study. Besides the
changes in the mineral dust emission scheme
already addressed above, further changes in the
model include in particular modified aerosol-
cloud interactions (Tegen et al., 2019). Due to the
full coupling of HAM2.3 to ECHAM6.3, all those
changes have eventually an effect on regional,
and thus global, dust emissions. Since the aim of
our study is completely different from Stanelle et
al. (2014), a thorough comparison between
results obtained with the outdated model setup
from 2014 and our new setup is beyond the scope
of our study. | According to our response, we made no further changes in our manuscript. |
| 261: Among the model factors
affecting dust emissions surely there is
also the vegetation cover, here
simulated thanks to a dynamic
vegetation model. I would suggest
adding a panel showing a map of the
vegetation fraction, or anyway a
vegetation-related variable that
closely resembles the way vegetation
affects dust emissions in the model.                                                                                                                                                                                                                                                                                                                                                                                                                                                                                           | Thank you for your suggestion, the dust
emissions are indeed affected by the dynamic
vegetation model. Please find in the following
two maps showing the simulated deserted
fraction of each grid box for PI and LGM as an
addition to Figure 2.                                                                                                                                                                                                                                                                                                                                                                                                                                                                                                                                                                                                                                                                                                                                                                                                                                                                                                                                                                                                                                                                                                                                                                                                                                                                                                                                                                                  | We included both subplots in Fig. 2.                                     |

| 283: The observational data used for
figure 3 do not appear to correspond
to the original DIRTMAP dataset (i.e.
Figure 8 in Kohfeld and Harrison,
2001). Please make sure that you add
a reference corresponding to the
actual version of the dataset you used,
and specify whether additional data
were included.                                                                                                                                                                                                                                                                                                                                                                                                                                                                                                                                                                  | Thank you for the hint! The correct reference is:
K.E. Kohfeld, R.M. Graham, A.M. de Boer, L.C.
Sime, E.W. Wolff, C. Le Quéré, L. Bopp: Southern
Hemisphere westerly wind changes during the
Last Glacial Maximum: paleo-data synthesis,
Quaternary Science Reviews, Volume 68, 2013.
No additional data were included. We will correct
the reference in our revised manuscript.                                                                                                                                                                                                                                                                                                                                                                                                                                                                                                                          | We corrected the according reference in line 313.                                     |
|-------------------------------------------------------------------------------------------------------------------------------------------------------------------------------------------------------------------------------------------------------------------------------------------------------------------------------------------------------------------------------------------------------------------------------------------------------------------------------------------------------------------------------------------------------------------------------------------------------------------------------------------------------------------------------------------------------------------------------------------------------------------------------------------------------------------------------------------------------------------------------------------------------------|--------------------------------------------------------------------------------------------------------------------------------------------------------------------------------------------------------------------------------------------------------------------------------------------------------------------------------------------------------------------------------------------------------------------------------------------------------------------------------------------------------------------------------------------------------------------------------------------------------------------------------------------------------------------------------------------------------------------------------------------------------------------------------------------------------------------------------------------------------------------------------------------------------------------------------|---------------------------------------------------------------------------------------|
| 283-315: Several data points in the
Southern Ocean appear to be south of
the Polar front, which should raise a
flag about non-aeolian contributions
to the terrigenous fraction of the
sediment, and therefore the
opportunity to use these data for a
robust estimation of dust mass
accumulation rates (e.g. Kohfeld and
Harrison, 2001).                                                                                                                                                                                                                                                                                                                                                                                                                                                                                                                                      | Although data from "[] marine sites that have
been flagged because they are located within
zones of thick nepheloid layers and ice-rafted
detritus, which can contaminate aeolian signals
[]" had already been excluded from the dataset
we use for comparison (Kohfeld et al., 2013), we
agree that the reconstructed detrital flux
estimates based on changes in 232 Th might still
contain non-aeolian contributions from glacier
erosion and riverine input, which are not
considered in our model. Additionally, it should
be taken into account that we compare
(simulated) aeolian dust deposition fluxes onto
the ocean surface to marine sediment data, i.e.
also any horizontal dust transport processes in
the ocean during sedimentation are not
considered. We will point this uncertainty out in
the discussion section in our revised manuscript. | We added the according aspects in the lines 336-348.                                  |
| 352-356: There is a substantial difference in the experimental design of Albani et al. (2012 and 2014) and this work; here it appears that the amount and proportions of dust from different sources result only from the model itself (and indirectly the regional tuning on dust emissions made on present day conditions, apparently), whereas the cited work explicitly used regional tuning also for the LGM, in a data-assimilation fashion, in order to obtain a match on dust amounts, LGM/interglacial ratio, as well as source mix based on geochemical fingerprinting on Antarctic ice core samples (e.g. Delmonte et al., 2010). In other words, one could say that the CAM3 results that you mention indicate a dominance of South American dust because ice core data suggest just that, of course under the assumption that simulated transport and deposition can be considered reasonable. | We agree! Albani et al. (2012) found the
dominance of South American dust only because
they tuned the dust emissions in their simulations
"for each macro-area [] a posteriori by applying
a factor yielding the best fit between the
simulated and observed LGM and current
deposition rates []" and is thus not suitable to
be used as a reference indicating contradicting
model results compared to our simulations. We
will adjust our argumentation accordingly in our
revised manuscript.                                                                                                                                                                                                                                                                                                                                                                                                 | We adjusted our argumentation
accordingly in the lines 399-403.                    |
| 352-368: Based on my previous
comment, I would recommend that a
more thorough discussion is carried
out considering also the available data
on dust provenance. It is indeed very
important that you explain your
results based on the modeled
processes, as you did, but I believe
that they should also be put more in
perspective by comparing them to
observational evidence, also for this
particular aspect (which by the way                                                                                                                                                                                                                                                                                                                                                                                                                                        | We will point out more clearly in our revised
manuscript that our model results are not
intended to question the geochemical data
regarding the provenance of dust found in
Antarctic ice cores.
Additionally, we will include in the discussion
section that the reconstructed dust fluxes used in
our study for comparison with our simulation
results (DIRTMAP, Kohfeld et al. (2013)), which
are based on the assumption of relatively
constant proportions of 232 Th in continental                                                                                                                                                                                                                                                                                                                                                                                              | We pointed out to the data-model
discrepancy in the lines 336-348 and 399-
403. |

| you mention later on while discussing
the matter of size, and you also
acknowledge in the conclusions).                                                                                   | lithogenic materials, might be overestimated by
30–40 % in regions receiving fine-grained dust
from Patagonia and Australia (McGee et al.,
2015). The study of Trudgill et al. (2020) supports
our finding of Australia being the predominant
dust source during the LGM for dust deposited in
the SW Pacific, however, they also suggest based
on their grain-size analysis of sediment cores
from the Tasman Sea that these might contain
non-aeolian contributions, more precisely fluvial
sediments from New Zealand, which are not
considered by our model and might partly explain
the discrepancy between our model results and
the observational data.                                                                              |                                                                                             |
|-------------------------------------------------------------------------------------------------------------------------------------------------------------------------------------------------|------------------------------------------------------------------------------------------------------------------------------------------------------------------------------------------------------------------------------------------------------------------------------------------------------------------------------------------------------------------------------------------------------------------------------------------------------------------------------------------------------------------------------------------------------------------------------------------------------------------------------------------------------------------------------------------------------------------------------------------------------------------------------------|---------------------------------------------------------------------------------------------|
| 412-414: Is there a variability on size
distributions at the stage of dust
emissions in your model formulation?
I don't think so, so I'm a bit confused,
why would you expect that? | Ice core data from Greenland (Steffensen, 1997)
and Antarctica (e.g. Delmonte et al., 2004)
indicate the dust deposition of varying particle
size distributions during glacials compared to
interglacials. Since we also find a change in dust
particle size during the LGM compared to PI (in
particular over Antarctica), this formulation has
been chosen to point out to the reader that
although one might expect that the model
exhibits this change in particle size for physical
reasons and thus might yield a possible
explanation for the according observational data,
it is caused for a different reason. Considering the
confusion this formulation has apparently
caused, we will rephrase this sentence
accordingly. | We rephrased this sentence accordingly in
the lines 478-482.                             |
| 472-474: Where does this come from?
This aspect is not shown or discussed
anywhere in the text.                                                                                           | We agree! It was not mentioned at an earlier
point in the text. However, this were the findings
of Stanelle et al. (2014) using an older version of
the model, so it can be considered very likely that
the same findings can be attributed to the same
causes. We will include the according reference
in our revised manuscript.                                                                                                                                                                                                                                                                                                                                                                                                                               | We removed the according sentence
because it is irrelevant in the scope of our
study. |
| 478-479: I would suggest adding two
lines bracketing the +/- 1 order of
magnitude in the scatterplots of
Figure 3, for a clearer reading.                                              | Thank you for this suggestion, please find below
the accordingly adjusted scatterplots.
(b) Dust deposition - PI 1850-1879
(c) Pacific SO
Antarctica
10 -0
10 -1
(d) Dust deposition [gm -2 yr -1 ]
(d) Dust deposition - LGM 21kyr BP                                                                                                                                                                                                                                             | We updated the according subplots in Fig.
3.                                             |

|                                                                                                                                                                              | (f) Dust deposition ratio - LGM/PI                                                                                                                                                                                                                                                                                                                                                                                                             |                                                                                                                                                                                                                                 |
|------------------------------------------------------------------------------------------------------------------------------------------------------------------------------|------------------------------------------------------------------------------------------------------------------------------------------------------------------------------------------------------------------------------------------------------------------------------------------------------------------------------------------------------------------------------------------------------------------------------------------------|---------------------------------------------------------------------------------------------------------------------------------------------------------------------------------------------------------------------------------|
|                                                                                                                                                                              | 10 2

0                                                                                                                                                                                                                                                                                                                                                               |                                                                                                                                                                                                                                 |
| 500-504: I would recommend that
these considerations on the chosen
boundary conditions are also reported
in the methods and/or results
sections, as appropriate. | We agree! We will include the considerations
about the potential influence of the prescribed
sea surface temperatures on our simulation
results already in the discussion in section 3.2.                                                                                                                                                                                                                                             | We included these considerations in the lines 264-266.                                                                                                                                                                          |
| 466-504: I would suggest enriching a
bit the conclusion section with
references to the literature, where
appropriate.                                               | Since we do not bring up new aspects to the discussion in our final paragraph 4. Conclusions, in particular after moving the considerations about the potential influence of the prescribed sea surface temperatures on our simulation results already up to discussion section 3.2, we tend to not give any references in the conclusion section at all because those relevant for our paper are already mentioned in the discussion section. | Since we enriched the discussion section
of our manuscript by several aspects
including the according references in the
scope of the revision process, we did not
give any references in the conclusion
section. |

**Referee's references:**

Balkanski, Y., Schulz, M., Claquin, T., & Guibert, S. (2007). Reevaluation of mineral aerosol radiative forcings suggests a better agreement with satellite and AERONET data. Atmospheric Chemistry and Physics, 7, 81–95. https://doi.org/10.5194/acpâ7â81â2007

Dufresne, J. L., Gautier, C., Ricchiazzi, P., & Fouquart, Y. (2002). Longwave scattering effects of mineral aerosols. Journal of the Atmospheric Sciences, 59, 1959–1966. https://doi.org/10.1175/1520â0469(2002)059<1959:LSEOMA>2.0.CO;2

Delmonte, B., Andersson, P., Schöberg, H., Hansson, M., Petit, J. R., Delmas, R., Gaiero, D. M., Maggi, V., and Frezzotti, M.: Geographic provenance of aeolian dust in East Antarctica during Pleistocene glaciations: preliminary results from Talos Dome and comparison with East Antarctic and new Andean ice core data, Quat. Sci. Rev., 29, 256– 264, doi:10.1016/j.quascirev.2009.05.010, 2010

**Author's references:**

Albani, S., Mahowald, N. M., Delmonte, B., Maggi, V., and Winckler, G.: Comparing modeled and observed changes in mineral dust transport and deposition to Antarctica between the Last Glacial Maximum and current climates, Clim. Dyn., 38, 1731–1755, https://doi.org/10.1007/s00382-011-1139-5, 2012.

Balkanski, Y., Schulz, M., Claquin, T., and Guibert, S.: Reevaluation of Mineral aerosol radiative forcings suggests a better agreement with satellite and AERONET data, Atmospheric Chem. Phys., 7, 81–95, https://doi.org/10.5194/acp-7-81-2007, 2007.

Boucher, O.: Atmospheric Aerosols: Properties and Climate Impacts, Springer Netherlands, https://doi.org/10.1007/978-94-017-9649-1, 2015.

Delmonte, B., Petit, J.-R., Andersen, K., Basile-Doelsch, I., Maggi, V., and Lipenkov, V.: Dust size evidence for opposite regional atmospheric circulation changes over East Antarctica during the last climatic transition, Clim. Dyn., 23, 427–438, https://doi.org/10.1007/s00382-004-0450-9, 2004.

Dufresne, J.-L., Gautier, C., Ricchiazzi, P., and Fouquart, Y.: Longwave Scattering Effects of Mineral Aerosols, J. Atmospheric Sci., 59, 1959–1966, https://doi.org/10.1175/1520-0469(2002)059<1959:LSEOMA>2.0.CO;2, 2002.

Kohfeld, K. E., Graham, R. M., de Boer, A. M., Sime, L. C., Wolff, E. W., Le Quéré, C., and Bopp, L.: Southern Hemisphere westerly wind changes during the Last Glacial Maximum: paleo-data synthesis, Quat. Sci. Rev., 68, 76–95, https://doi.org/10.1016/j.quascirev.2013.01.017, 2013.

McGee, D., Winckler, G., Borunda, A., Serno, S., Anderson, R., Recasens, C., Bory, A., Gaiero, D., Jaccard, S., Kaplan, M., McManus, J., Revel, M., and Sun, Y.: Tracking eolian dust with helium and thorium: Impacts of grain size and provenance, Geochim. Cosmochim. Acta, 175, https://doi.org/10.1016/j.gca.2015.11.023, 2015.

Steffensen, J. P.: The size distribution of microparticles from selected segments of the Greenland Ice Core Project ice core representing different climatic periods, J. Geophys. Res. Oceans, 102, 26755–26763, https://doi.org/10.1029/97JC01490, 1997.

**Referee 2:**

| Comment by referee 2                                                                                                                                                                                                                                                                                                                                                                                                                                                                                                                                                                                                                                                                                                                                                                                                                                                                                                                                                                                                                                                                                                                                                      | Reply by authors                                                                                                                                                                                                                                                                                                                                                                                                                                                                                                                                                                                                                                                                                                                                                                                                                                                                                                                                                                                                                                                                                                                                                                                                                                                                                                                                                                                                                                                                                                                                                                                                                                                                                        | Changes in the manuscript                                                                                                                                                                                                                                                            |  |
|---------------------------------------------------------------------------------------------------------------------------------------------------------------------------------------------------------------------------------------------------------------------------------------------------------------------------------------------------------------------------------------------------------------------------------------------------------------------------------------------------------------------------------------------------------------------------------------------------------------------------------------------------------------------------------------------------------------------------------------------------------------------------------------------------------------------------------------------------------------------------------------------------------------------------------------------------------------------------------------------------------------------------------------------------------------------------------------------------------------------------------------------------------------------------|---------------------------------------------------------------------------------------------------------------------------------------------------------------------------------------------------------------------------------------------------------------------------------------------------------------------------------------------------------------------------------------------------------------------------------------------------------------------------------------------------------------------------------------------------------------------------------------------------------------------------------------------------------------------------------------------------------------------------------------------------------------------------------------------------------------------------------------------------------------------------------------------------------------------------------------------------------------------------------------------------------------------------------------------------------------------------------------------------------------------------------------------------------------------------------------------------------------------------------------------------------------------------------------------------------------------------------------------------------------------------------------------------------------------------------------------------------------------------------------------------------------------------------------------------------------------------------------------------------------------------------------------------------------------------------------------------------|--------------------------------------------------------------------------------------------------------------------------------------------------------------------------------------------------------------------------------------------------------------------------------------|--|
| Primary comments                                                                                                                                                                                                                                                                                                                                                                                                                                                                                                                                                                                                                                                                                                                                                                                                                                                                                                                                                                                                                                                                                                                                                          |                                                                                                                                                                                                                                                                                                                                                                                                                                                                                                                                                                                                                                                                                                                                                                                                                                                                                                                                                                                                                                                                                                                                                                                                                                                                                                                                                                                                                                                                                                                                                                                                                                                                                                         |                                                                                                                                                                                                                                                                                      |  |
| The paper would benefit from more
time in the introduction and conclusion
spent reviewing what is known and the
disagreements regarding dust
provenance to Antarctic and the
Southern Ocean. I was pleased when
the authors brought up many of the
studies when discussing their results,
but I felt the bigger picture was
somewhat overlooked. Specifically,
there are conflicting studies regarding
whether South America or Australia is
the primary source of dust (and for that
matter how much is contributed by
Antarctic sources), and the relative role
of dust source strength and transport
efficiency. I think the authors have
room here to set up and then answer
some questions about how these
discrepancies can be resolved by
considering the time-varying relative
strength of sources, the transport
efficiency, and the spatial distribution
of their influence. I think much of this
information is already contained in the
paper, but an explicit consideration of
the debate would be valuable. One
additional source to consider is Markle,
et al. (2018). | Since the provenance studies for dust in the
Southern Hemisphere are an important part of our
study, we agree and will include a brief overview
on the conflicting studies regarding whether South
America or Australia is the primary source of dust
already into our introduction section. As written in
our reply to Eric Wolff's comment on our
manuscript 1 , we will not be able to give an
ultimate answer on the question whether changes
in source strength or transport efficiency
eventually led to the observed increase in mineral
dust aerosol concentration during the LGM found
in marine sediments and Antarctic ice cores.
Considering the fact that more than 90% of the
dust deposition in the Southern Ocean region
occurs due to precipitation in our model, we agree
with Markle et al. (2018) that "precipitation in the
mid-latitudes is the principal barrier to aerosol
reaching the poles". However, we disagree that
changes in the hydrologic cycle are the primary
driver since we also find substantial and required
changes in source strength by a factor of 16 for
both Patagonia and Australia, which we can trace
back well to changes in vegetation, wind speed, soil
moisture and extension of the source areas.
Moreover, the authors find the best agreement
between their modeling results and data "at multi-
centennial and longer timescales", while our study
captures only a 30-year period under LGM
conditions and thus considers in particular
processes on much shorter timescales. We will
include those aspects in our revised manuscript. | We included a brief overview on the
conflicting dust provenance studies in
the introduction section in the lines 66-
79.
We combined our results with the
ongoing debate on the relative role of
source strength and transport efficiency
in the lines 431-450. |  |
| What about New Zealand? I was
surprised that the LGM simulations
don't seem to include an expanded
dust source from the exposed
continental shelf around New Zealand,
nor any discussion of it as a dust source
during that period. Neff, et al. (2015)
and Koffman, et al. (2021) would be
relevant to this discussion.                                                                                                                                                                                                                                                                                                                                                                                                                                                                                                                                                                                                                                                                                                                                                                                                                          | We are aware of the ongoing discussions on New Zealand as a potential dust source especially for the South Pacific during the last LGM (e.g. Lamy et al., 2014). Our model yields annual dust emissions of less than 1 Gg yr -1 from New Zealand during the LGM, which is effectively negligible compared to the simulated emissions of 748 Tg yr -1 (Australia) and 36 Tg yr -1 (Patagonia). The low emissions in our model also explain why New Zealand appears to not represent a dust source at all during the LGM in Fig. 1b ( paper ).
One reason to consider is that New Zealand's geographical expanse is rather small and thus only marginally captured by our model running in the spatial resolution of T63, which corresponds to a grid box size of approximately 180 km (Sidorenko et al., 2015). Consequently, New Zealand's source strength and the according contribution to the dust deposition in the Pacific sector of the Southern Ocean during the LGM (Koffman et al., 2021) might be underestimated in our model. The study of Neff and Bertler (2015) uses a trajectory modeling approach for the years 1979 to 2013 based on reanalysis datasets for the according pressure                                                                                                                                                                                                                                                                                                                                                                                                                                                         | We added the according information in
the lines 367-372.                                                                                                                                                                                                                          |  |

 ${}^1\,https://cp.copernicus.org/preprints/cp-2021-73/cp-2021-73-AC1-supplement.pdf$

|                                                                                                                                                                                                                                                                                | fields. Additionally, the authors mention in their
study that they investigate solely the trajectories
without taking into account emission and
deposition processes, which does not enable us to
compare our simulation results for the LGM to the
results of their study.
However, we agree that our findings concerning
New Zealand's role as a dust source during the
LGM should be included in our study and we will do
so in our revised manuscript.                                                                                                                                                                                                                                                                                                                                                                                                                                                                                                                                                                                                                                                                                                                                                                                                                                                                                                                                                                                                                         |                                                                                                                                                                                                                                                                                                                                                                 |
|--------------------------------------------------------------------------------------------------------------------------------------------------------------------------------------------------------------------------------------------------------------------------------|---------------------------------------------------------------------------------------------------------------------------------------------------------------------------------------------------------------------------------------------------------------------------------------------------------------------------------------------------------------------------------------------------------------------------------------------------------------------------------------------------------------------------------------------------------------------------------------------------------------------------------------------------------------------------------------------------------------------------------------------------------------------------------------------------------------------------------------------------------------------------------------------------------------------------------------------------------------------------------------------------------------------------------------------------------------------------------------------------------------------------------------------------------------------------------------------------------------------------------------------------------------------------------------------------------------------------------------------------------------------------------------------------------------------------------------------------------------------------------------------------------------|-----------------------------------------------------------------------------------------------------------------------------------------------------------------------------------------------------------------------------------------------------------------------------------------------------------------------------------------------------------------|
| The authors cover many of the
modeling studies of dust transport to
the Antarctic in their discussion of
provenance, but there are also studies
that take an isotopic approach that
should be discussed as well. Wengler et
al. (2019), McGee et al. (2016). | The study of Wengler et al. (2019) provides
lithogenic flux data only for the Holocene, stating
that surface sediments near New Zealand "most
likely indicate a combination of Australian dust and
riverine input from New Zealand". Since our model
only considers aeolian dust fluxes, we cannot
compare our simulation results for PI and LGM.
However, we will include in our revised manuscript
that the reconstructed dust fluxes used in our
study for comparison with our simulation results
(DIRTMAP, Kohfeld et al. (2013)), which are based
on the assumption of relatively constant
proportions of 232 Th in continental lithogenic
materials, might be overestimated by 30–40 % in
regions receiving fine-grained dust from Patagonia
and Australia (McGee et al., 2015).                                                                                                                                                                                                                                                                                                                                                                                                                                                                                                                                                                                                                                                                | We considered studies on dust
provenance based on an isotopic
approach in the lines 66-79, 336-348
and 399-403.                                                                                                                                                                                                                                        |
| Minor comments                                                                                                                                                                                                                                                                 |                                                                                                                                                                                                                                                                                                                                                                                                                                                                                                                                                                                                                                                                                                                                                                                                                                                                                                                                                                                                                                                                                                                                                                                                                                                                                                                                                                                                                                                                                                               |                                                                                                                                                                                                                                                                                                                                                                 |
| How is land tiling / vegetation coverage
determined for the newly exposed
continental shelf? Essentially, is the
newly exposed land always a dust
source, or can it become vegetated?                                                                              | As mentioned in our manuscript, we perform a restart for our LGM experiments using restart files, which represent a dynamic equilibrium of our model for the according topographic, vegetation and climate conditions obtained after several hundred years of simulation (line 172). Generally, the land fraction of each grid cell can become vegetated according to the rules and equations for dynamic vegetation implemented in the land surface and vegetation model JSBACH, which are described in detail in Reick et al. (2013), paragraph 3 "Natural Land Cover Change" . In short, the process of increasing / decreasing vegetation for each grid cell depends on several factors like meteorological conditions (temperature, precipitation, humidity,), competition among the different plant functional types (PFT), the respective time constants for growth and lifetime for each PFT etc One of the leading principals is the so-called universal presence , i.e. seeds of each PFT are always on the land fraction in each grid cell available and the factors mentioned before determine the respective proportion of each PFT per grid cell. The land fraction. The latter has been used by (Stanelle et al., 2014) to determine the area for dust emissions in each grid cell, which are additionally influenced by several meteorological factors like wind speed, soil moisture etc The following maps show the desert fraction for each grid cell for PI and LGM. | Since the reference to the dynamic
vegetation model JSBACH is given in line
127 and a detailed description of the
model would be beyond the scope of
our study, we did not include further
background information.
However, we included the according
subplots on vegetation in Fig. 2, showing
the desert fraction for each grid cell. |

|                                                                                                                                                                                                                                                                                                                 | (g) V                                                                                                                                                                                                                                                                                                                                                                                                                                                                                                                                                                                                                                                                                                                                                                                                                                                                                  | egetation - PI 1                                                                                                                                                                         | 850-1879                                                                                                                                                             |                                                                                                                                                                    |                                                                                                                        |
|-----------------------------------------------------------------------------------------------------------------------------------------------------------------------------------------------------------------------------------------------------------------------------------------------------------------|----------------------------------------------------------------------------------------------------------------------------------------------------------------------------------------------------------------------------------------------------------------------------------------------------------------------------------------------------------------------------------------------------------------------------------------------------------------------------------------------------------------------------------------------------------------------------------------------------------------------------------------------------------------------------------------------------------------------------------------------------------------------------------------------------------------------------------------------------------------------------------------|------------------------------------------------------------------------------------------------------------------------------------------------------------------------------------------|----------------------------------------------------------------------------------------------------------------------------------------------------------------------|--------------------------------------------------------------------------------------------------------------------------------------------------------------------|------------------------------------------------------------------------------------------------------------------------|
|                                                                                                                                                                                                                                                                                                                 |                                                                                                                                                                                                                                                                                                                                                                                                                                                                                                                                                                                                                                                                                                                                                                                                                                                                                        |                                                                                                                                                                                          |                                                                                                                                                                      |                                                                                                                                                                    |                                                                                                                        |
| Since additional land area during the                                                                                                                                                                                                                                                                           | Plaza find the                                                                                                                                                                                                                                                                                                                                                                                                                                                                                                                                                                                                                                                                                                                                                                                                                                                                         | Desert fract                                                                                                                                                                             | ion                                                                                                                                                                  | on in the                                                                                                                                                          | We included the according table as                                                                                     |
| Since additional land area during the
LGM is credited as one of the causes of
increased dust emission, I would like
some information, similar to the
reported wind and precip changes, that
tells me how much additional land
there is in each region (and possibly
some of how much a the | Please find the
following table.
Addition
land are
LGM
[Mio.
km²]                                                                                                                                                                                                                                                                                                                                                                                                                                                                                                                                                                                                                                                                                                                                                                                                    | according
al Dust
a Emission
[Tg yr -1 ]
PI 1850-
1879                                                                                                         | Dust
Emission
[Tg yr -1 ]
LGM
21kyr BP                                                                                                        | DN IN the
Dust
Emission
[Tg yr -1 ]
on
additional
land
areas                                                                       | We included the according table as
Table 4 in our manuscript and discussed
certain aspects in the lines 284-287. |
| dust is being created from this new                                                                                                                                                                                                                                                                             | Globally 19.5                                                                                                                                                                                                                                                                                                                                                                                                                                                                                                                                                                                                                                                                                                                                                                                                                                                                          | 923                                                                                                                                                                                      | 5159                                                                                                                                                                 | 230                                                                                                                                                                |                                                                                                                        |
| land).                                                                                                                                                                                                                                                                                                          | Australia 1.8                                                                                                                                                                                                                                                                                                                                                                                                                                                                                                                                                                                                                                                                                                                                                                                                                                                                          | 47                                                                                                                                                                                       | 748                                                                                                                                                                  | 92                                                                                                                                                                 |                                                                                                                        |
|                                                                                                                                                                                                                                                                                                                 | Southern 0.04                                                                                                                                                                                                                                                                                                                                                                                                                                                                                                                                                                                                                                                                                                                                                                                                                                                                          | 12                                                                                                                                                                                       | 63                                                                                                                                                                   | 5                                                                                                                                                                  |                                                                                                                        |
|                                                                                                                                                                                                                                                                                                                 | Patagonia 0.8                                                                                                                                                                                                                                                                                                                                                                                                                                                                                                                                                                                                                                                                                                                                                                                                                                                                          | 2.3                                                                                                                                                                                      | 36                                                                                                                                                                   | 29                                                                                                                                                                 |                                                                                                                        |
|                                                                                                                                                                                                                                                                                                                 | Globally as well as
the additional lan
lower sea level du
small proportion d
Consequently, the
are changes in ve
meteorological fa
patterns and wind
the additional la
drylands (see v
substantially to th
can also be seen in                                                                                                                                                                                                                                                                                                                                                                                                                                                                                                                                                                                                                        | in Australia
d areas as a
uring the LG
to the addit
e main reas
egetation (s
ctors, for in
d speed. In
nd area, r
regetation
e absolute o
n Fig. 1b (po | and South
a conseque
SM contrib
ionally em
ons for th
ee maps a
nstance pr
Patagonia
mainly con
maps), c
dust emissi
sper ). | hern Africa,
ence of the
bute only a
litted dust.
he increase
above) and
recipitation
, however,
nsisting of
contributes
ions, which |                                                                                                                        |
| In Figure 3, why are the simulated dust
deposition values so stratified? The
observations appear continuous across
a couple orders of magnitude, while
the simulated deposition values appear
to form horizontal lines.                                                                          | The stratification of the simulated dust deposition
values indicate that the model is not able to
capture the observed variation for both PI and
LGM, in particular in the Pacific / Pacific SO region.
Since the measurement locations are rather close
to each other, the discrepancy might be caused by
a shortcoming in the representation of the
deposition process on a small scale in our model.
However, it should be taken into account that we
compare aeolian dust deposition fluxes onto the
ocean surface to marine sediment data, i.e. aside
from potential shortcomings in our model,
horizontal dust transport processes in the ocean
during sedimentation as well as dust flux
contributions due to glacier erosion and fluvial
inputs are not considered and might play a crucial
role (e.g. Trudgill et al. (2020)) |                                                                                                                                                                                          |                                                                                                                                                                      |                                                                                                                                                                    | We discussed the data-model discrepancy in the lines 336-348.                                                          |
| Continuous colorbars on log plots are
difficult to accurately interpret. The
maps in Figures 3 and 1 would be much
easier to read if the colorbars had                                                                                                                                                 | Thank you for th
accordingly adjust                                                                                                                                                                                                                                                                                                                                                                                                                                                                                                                                                                                                                                                                                                                                                                                                                                                 | nis suggesti
ed plots (Fig                                                                                                                                                            | ion, pleaso
g. 3a, c, e)                                                                                                                                          | e find the
below.                                                                                                                                               | We updated the according subplots in Fig. 3.                                                                           |

---

## Author Response (AR2)

**Detailed replies to both anonymous referees comments on**

Krätschmer, S., van der Does, M., Lamy, F., Lohmann, G., Völker, C., and Werner, M.: *Simulating glacial dust changes in the Southern Hemisphere using ECHAM6.3-HAM2.3*, Clim. Past Discuss. [preprint], https://doi.org/10.5194/cp-2021-73, in review, 2021.

**2nd Revision: Technical Corrections**

**Referee 1:**

| Comment by referee 1 | Reply by authors | Changes in the manuscript |
|---|---|---|
| Concerning my comment on (old) lines 213-214, just to clarify, I was not encouraging the authors to compare their results with (Stanelle et al., 2014), but rather to show the comparison of their model setup in current climate conditions with (Huneeus et al., 2011), or to provide a reference that does that with the same model version and setup used here. | Thank you for the clarification. Please find below the comparison of our simulation results for present-day dust deposition at various sites to observed data provided by Huneeus et al. (2011). Generally, our model results match well with the observational data. In particular in the West Pacific we find an improvement compared to the previous model version used by Stanelle et al. (2014).

 | We included the two figures in the Supplement and added an according reference in the lines 236-237. |
| 305: It looks like it should be "Fig. 2g and h". | Thank you for the hint, we will correct it in a revised version of our manuscript. | We have corrected the reference in line 306. |
| 313: I am not sure that the citation to (Kohfeld et al., 2013) should be referred to as "DIRTMAP"; perhaps you could say simply e.g. "the compilation of dust deposition from Kohfeld et al. (2013)" | We will rephrase the sentence as follows:

"We use the compilation of dust deposition data from Kohfeld et al. (2013) for a comparison […]" | We have rephrased the sentence accordingly in line 314. |
| 431-433: Perhaps the improved discussion of section 3.2.4 would benefit from the results on the influence of deposition mechanisms on dust size not only over the Southern Ocean, but also above Antarctica (e.g. Albani et al., 2012). | Thank you for the suggestion, we will include the findings of Albani et al. (2012) on the influence of wet and dry deposition on the observed spatial variation in particle size in Antarctica during the LGM. | We have included the according aspects in our discussion in the lines 475-481. |

**Referee's references:**

Albani, S., Mahowald, N. M., Delmonte, B., Maggi, V., and Winckler, G.: Comparing modeled and observed changes in mineral dust transport and deposition to Antarctica between the Last Glacial Maximum and current climates, 38, 1731–1755, https://doi.org/10.1007/s00382-011-1139-5, 2012.

Huneeus, N., Schulz, M., Balkanski, Y., Griesfeller, J., Prospero, J., Kinne, S., Bauer, S., Boucher, O., Chin, M., Dentener, F., Diehl, T., Easter, R., Fillmore, D., Ghan, S., Ginoux, P., Grini, A., Horowitz, L., Koch, D., Krol, M. C., Landing, W., Liu, X., Mahowald, N., Miller, R., Morcrette, J.-J., Myhre, G., Penner, J., Perlwitz, J., Stier, P., Takemura, T., and Zender, C. S.: Global dust model intercomparison in AeroCom phase I, 11, 7781–7816, https://doi.org/10.5194/acp-11-7781-2011, 2011.

Kohfeld, K. E., Graham, R. M., de Boer, A. M., Sime, L. C., Wolff, E. W., Le Quéré, C., and Bopp, L.: Southern Hemisphere westerly wind changes during the Last Glacial Maximum: paleo-data synthesis, 68, 76–95, https://doi.org/10.1016/j.quascirev.2013.01.017, 2013.

Stanelle, T., Bey, I., Raddatz, T., Reick, C., and Tegen, I.: Anthropogenically induced changes in twentieth century mineral dust burden and the associated impact on radiative forcing, 119, 13,526-13,546, https://doi.org/10.1002/2014JD022062, 2014.

**Author's references:**

Albani, S., Mahowald, N. M., Delmonte, B., Maggi, V., and Winckler, G.: Comparing modeled and observed changes in mineral dust transport and deposition to Antarctica between the Last Glacial Maximum and current climates, Clim. Dyn., 38, 1731–1755, https://doi.org/10.1007/s00382-011-1139-5, 2012.

Huneeus, N., Schulz, M., Balkanski, Y., Griesfeller, J., Prospero, J., Kinne, S., Bauer, S., Boucher, O., Chin, M., Dentener, F., Diehl, T., Easter, R., Fillmore, D., Ghan, S., Ginoux, P., Grini, A., Horowitz, L., Koch, D., Krol, M. C., Landing, W., Liu, X., Mahowald, N., Miller, R., Morcrette, J.-J., Myhre, G., Penner, J., Perlwitz, J., Stier, P., Takemura, T., and Zender, C. S.: Global dust model intercomparison in AeroCom phase I, Atmospheric Chem. Phys., 11, 7781–7816, https://doi.org/10.5194/acp-11-7781-2011, 2011.

Kohfeld, K. E., Graham, R. M., de Boer, A. M., Sime, L. C., Wolff, E. W., Le Quéré, C., and Bopp, L.: Southern Hemisphere westerly wind changes during the Last Glacial Maximum: paleo-data synthesis, Quat. Sci. Rev., 68, 76–95, https://doi.org/10.1016/j.quascirev.2013.01.017, 2013.

Stanelle, T., Bey, I., Raddatz, T., Reick, C., and Tegen, I.: Anthropogenically induced changes in twentieth century mineral dust burden and the associated impact on radiative forcing, J. Geophys. Res. Atmospheres, 119, 13,526-13,546, https://doi.org/10.1002/2014JD022062, 2014.

**Referee 2:**

Referee 2 did not suggest any further revisions and recommended publication in its current form.